# Genomic and fitness consequences of a near-extinction event in the northern elephant seal

Joseph I. Hoffman [1,2,3,4,5] ✉, David L. J. Vendrami [1,3,5], Kosmas Hench [1,3], Rebecca S. Chen [1,3], Martin A. Stoffel [1,14], Marty Kardos [6], William Amos [7], Jörn Kalinowski [8], Daniel Rickert [9], Karl Köhrer [9], Thorsten Wachtmeister [9], Mike E. Goebel [10], Carolina A. Bonin [11], Frances M. D. Gulland [12] & Kanchon K. Dasmahapatra [13]

Understanding the genetic and fitness consequences of anthropogenic bottlenecks is crucial for biodiversity conservation. However, studies of bottlenecked populations combining genomic approaches with fitness data are rare. Theory predicts that severe bottlenecks deplete genetic diversity, exacerbate inbreeding depression and decrease population viability. However, actual outcomes are complex and depend on how a species' unique demography affects its genetic load. We used population genetic and veterinary pathology data, demographic modelling, whole-genome resequencing and forward genetic simulations to investigate the genomic and fitness consequences of a near-extinction event in the northern elephant seal. We found no evidence of inbreeding depression within the contemporary population for key fitness components, including body mass, blubber thickness and susceptibility to parasites and disease. However, we detected a genomic signature of a recent extreme bottleneck (effective population size = 6; 95% confidence interval = 5.0–7.5) that will have purged much of the genetic load, potentially leading to the lack of observed inbreeding depression in our study. Our results further suggest that deleterious genetic variation strongly impacted the post-bottleneck population dynamics of the northern elephant seal. Our study provides comprehensive empirical insights into the intricate dynamics underlying species-specific responses to anthropogenic bottlenecks.

Habitat destruction and overexploitation by humans have drastically decreased the abundance of numerous wild populations and driven some species to the brink of extinction[1–3]. These population bottlenecks impose both demographic and genetic threats to species persistence. Very small populations are vulnerable to extinction due to stochastic temporal variation in survival and reproduction[4], whereas severely bottlenecked populations are expected to rapidly lose genetic variation and the ability to adapt to future environmental changes[5], as well

as to quickly accumulate inbreeding and high-frequency deleterious alleles due to strong genetic drift[6]. However, in practice, the genetic and fitness outcomes of anthropogenic bottlenecks are complex and difficult to predict, partly because they depend on the specific demographic history of each species[7]. Unfortunately, these histories remain unknown for most organisms, making it challenging to comprehend the long-term effects of human-induced population bottlenecks on biodiversity and ecosystem health.

The genetic and fitness consequences of population bottlenecks also remain poorly understood due to the paucity of data on these effects in the wild and the complex and opposing effects of selection and genetic drift on deleterious genetic variation[8]. The constant input of deleterious alleles via mutation means that all species and populations carry a genetic load (decreased fitness due to the presence of deleterious alleles)[9,10]. Conceptually, this can be decomposed into the realized load and inbreeding load (otherwise known as the masked load)[8,11,12]. The realized load is the fraction of the total load that is expressed and which directly decreases the fitness of the population. It is determined by homozygous deleterious mutations and heterozygous deleterious mutations that are not fully recessive. It also includes deleterious mutations that have drifted to fixation (the drift load), which decrease the fitness of all individuals. The inbreeding load is the fraction of the total load that is masked in the heterozygous state and which causes inbreeding depression (that is, the decreased fitness of individuals with more closely related parents)[13]. While strong inbreeding depression has been detected in many wild populations[14], empirical results on the effects of inbreeding on population growth and viability are complex and often contradictory[15–18]. Regardless, the near-universal increase in fitness of small, isolated and declining populations following immigration[19] suggests that deleterious genetic variation often affects population dynamics.

Why do some populations appear to be threatened by genetic factors while others seem to be buffered against these effects? The widely varying effects of deleterious genetic variation on population dynamics are believed to arise partly due to the effects of demographic history on the genetic load[8,11,12], as well as on which fitness components are most affected[7]. Historically small populations are expected to have lower inbreeding loads because new deleterious mutations are often lost to strong genetic drift and the purging of partially recessive deleterious alleles exposed to natural selection via inbreeding[11,20]. However, the inefficiency of selection against weakly deleterious alleles in the face of strong genetic drift means that weakly deleterious alleles can easily become fixed in small populations, resulting in an elevated drift load compared with those of historically large populations[12,21]. Thus, populations with small historical effective population sizes ($N_e$) are expected to have lower inbreeding loads and therefore exhibit weaker inbreeding depression than populations with larger historical $N_e$, but they should also have a higher drift load and hence lower average intrinsic fitness[8,21,22].

The northern elephant seal (*Mirounga angustirostris*) provides a compelling opportunity to investigate the complex interplay between population declines, genetic load dynamics and fitness[23]. This iconic pinniped was abundant in the eighteenth century and was widely distributed along the Pacific coast of North America[24]. Extensive hunting by commercial sealers between 1810 and 1860 largely eliminated the species from most of its geographical range and it was considered extinct by the 1890s[25,26]. Fortunately, a small population survived on Guadalupe Island, which grew to around 350 seals by 1922 (ref. 27), when the northern elephant seal was protected by law. Although the subsequent recovery was initially slow, there followed over half a century of explosive population growth and range expansion[27–29], and by 2010 the estimated global population was around 225,000 animals[30]. Hence, the northern elephant seal is unusual among mammals in having experienced such a severe bottleneck followed by an unparalleled population increase[29,31,32,33]. The genetic and fitness consequences of this extreme bottleneck have been of major interest for half a century[34].

To investigate the genomic and fitness legacy of the severe anthropogenic bottleneck in northern elephant seals, we first used a suite of fitness measures and molecular genetic estimates of inbreeding to test for inbreeding depression. We then used population genomic data to evaluate the severity of the bottleneck and its likely impact on the genetic load. To validate our findings, we used whole-genome resequencing to compare the genomic landscape of deleterious variation in the northern elephant seal with that of its sister species, the southern elephant seal (*Mirounga leonina*), which did not experience a strong bottleneck[31]. Finally, we used forward genetic simulations to assess the likely impact of the bottleneck and purging of deleterious alleles on population recovery.

## Results and discussion

### Inbreeding depression

To test for inbreeding depression in northern elephant seals, we combined veterinary pathology data from 219 animals brought into The Marine Mammal Center in California for rehabilitation with molecular genetic data obtained from 22 microsatellites (Supplementary Table 1). A representative subset of 96 animals was then restriction-site associated DNA (RAD) sequenced to produce a quality-filtered dataset of 74 individuals genotyped at 15,051 single-nucleotide polymorphisms (SNPs) (see Methods). To test for population structure, we subjected both datasets to principal component analysis (PCA). No distinct genetic clusters were detected (Extended Data Fig. 1), suggesting that our samples originate from a genetically homogenous population.

To evaluate the ability of the molecular markers to capture variation in inbreeding, we calculated the two-locus identity disequilibrium statistic $g_2$ (ref. 35) separately for the microsatellites and SNPs. A clear signal of variation in inbreeding among individuals was detected, which was more readily resolved with the larger SNP dataset (Fig. 1a; 22 microsatellites: $g_2 = 0.012$; 95% confidence interval (CI) = −0.0014–0.0273; and 15,051 SNPs: $g_2 = 0.011$; 95% CI = 0.0032–0.0202), in line with theoretical expectations and previous empirical studies[36,37]. While the ability to capture variation in inbreeding is a prerequisite for detecting inbreeding depression with molecular markers[38], the power to detect inbreeding depression for different fitness components will depend on their underlying genetic architectures[39,40], as well as on the extent to which they are influenced by other sources of variation[41,42].

In large mammals that suffer negligible predation, fitness is predominantly determined by traits such as body size and condition (which impact longevity and reproductive success) and immune function (since this plays a major role in susceptibility to disease)[43–45]. We therefore tested for inbreeding depression for body mass and blubber thickness, which reflect nutritional status[46] and predict survival[47,48] and reproductive performance[48] in pinnipeds. We implemented separate Bayesian linear mixed models for each fitness component, fitting $z$-transformed standardized multilocus heterozygosity (sMLH) as a predictor variable and including sex and the month and year of admittance as random effects (see Methods for details). The 95% CIs of the posterior distributions of the standardized $\beta$ coefficients of sMLH overlapped zero for both the microsatellites and SNPs (Fig. 1b and Supplementary Table 2), indicating that inbreeding is not associated with either body mass or blubber thickness.

As described in the Methods, the animals were classified by an experienced marine mammal veterinarian (F.M.D.G.) into six categories, each representing a specific disease or condition that was the most likely cause of death: (1) helminth infection; (2) bacterial infection; (3) protozoan infection; (4) trauma; (5) malnutrition; and (6) congenital defects. We then tested for inbreeding depression for parasite and disease susceptibility by comparing levels of inbreeding among individuals assigned to these different categories. We implemented separate Bayesian generalized linear mixed models (GLMMs) for each category using a binomial response variable with 1 indicating that the respective disease or condition was the most likely cause of death of a given individual and 0 indicating that the individual was assigned to a different category. In all cases, the 95% CIs of the posterior distributions of the standardized $\beta$ coefficients of sMLH overlapped 0 for both the microsatellites and SNPs (Fig. 1c and Supplementary Table 3), suggesting that none of the categories differed in inbreeding relative to all of the other categories combined (Fig. 1d). We also found no effects of inbreeding in analyses where trauma was defined as a control category following

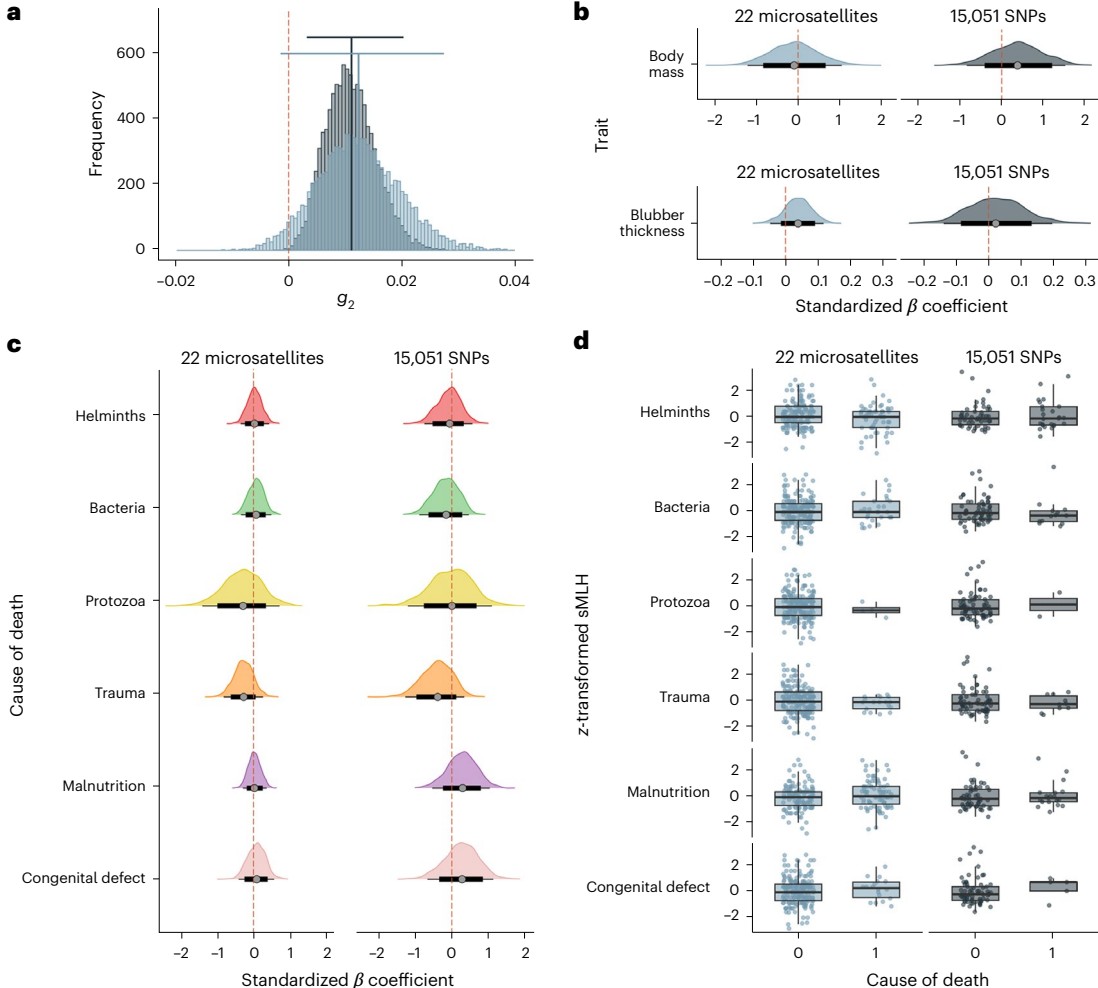

**Fig. 1 | Variance in inbreeding and the relationship between inbreeding and fitness components, including body mass, blubber thickness and susceptibility to parasites and disease, in northern elephant seals.**
**a**, Distribution of bootstrapped $g_2$ estimates obtained from 22 microsatellites genotyped in 219 individuals (light grey) and from 15,051 SNPs genotyped in 74 individuals (dark grey). The empirical $g_2$ values and their corresponding 95% CIs are depicted by vertical lines and horizontal bars, respectively.
**b**, Posterior distributions of the standardized $\beta$ coefficients of sMLH on body mass and blubber thickness for 22 microsatellites (light grey) and 15,051 SNPs (dark grey). The points represent the mean posterior estimates, the thick black lines represent 80% CIs and the thin black lines represent 95% CIs. **c**, Posterior

distributions of the standardized $\beta$ coefficients of sMLH on binary classifications of the most likely causes of death (see Methods for details). The three infectious disease categories (helminth, bacterial and protozoan infection) are shown in the top half of the plot. The points represent the mean posterior estimates, the thick black lines represent 80% CIs and the thin black lines represent 95% CIs.
**d**, Z-transformed sMLH values for each category, where 1 indicates that the respective disease or condition was the most likely cause of death of a given individual. Thick horizontal lines represent median z-transformed sMLH estimates, the lower and upper hinges correspond to the first and third quartiles, respectively, and the whiskers represent 1.5× the interquartile range.

Acevedo-Whitehouse et al.[49] (Supplementary Results and Discussion, Extended Data Fig. 2 and Supplementary Table 3).

These findings are in marked contrast with a previous study reporting strong associations between microsatellite heterozygosity and helminth and bacterial infection in California sea lions from the same rehabilitation centre using virtually identical protocols[49]. The primary differences between these studies lie with the species chosen and the level of genetic resolution, which is substantially higher in the current study. Consequently, we believe the two species differ in some key factor. One possibility is that northern elephant seals face fewer pathogenic threats, perhaps because some of their pathogens were not present in the remnant Guadalupe Island population and thus went extinct during the bottleneck. Alternatively, this species may have a much lower inbreeding load linked to disease susceptibility because the relevant alleles were purged or drifted to fixation as a result of the bottleneck. Elsewhere, heterozygosity has been linked to parasite infection in California sea lions[50], New Zealand sea lions[51] and harbour seals[37,52],

whereas more generally heterozygosity is associated with diverse life history traits in pinnipeds, from early survival[18] to lifetime reproductive success[23,53,54]. Hence, the lack of detectable inbreeding depression for parasite and disease susceptibility in northern elephant seals is conspicuous, particularly given the high resolution of our genomic data.

## Demographic reconstruction

We used the coalescent simulator fastsimcoal2 (ref. 55) to reconstruct the recent demographic history of the northern elephant seal using the site frequency spectrum (SFS) derived from all 96 RAD-sequenced individuals (see Methods for details). Three demographic models were compared (Extended Data Fig. 3). The first model included a recent bottleneck lasting for six generations (spanning 23–17 generations ago), corresponding to the known period of intensive harvesting. The second model included a recent bottleneck lasting for ten generations (spanning 23–13 generations ago), which included a subsequent period of lower-level hunting during which most of the remaining individuals

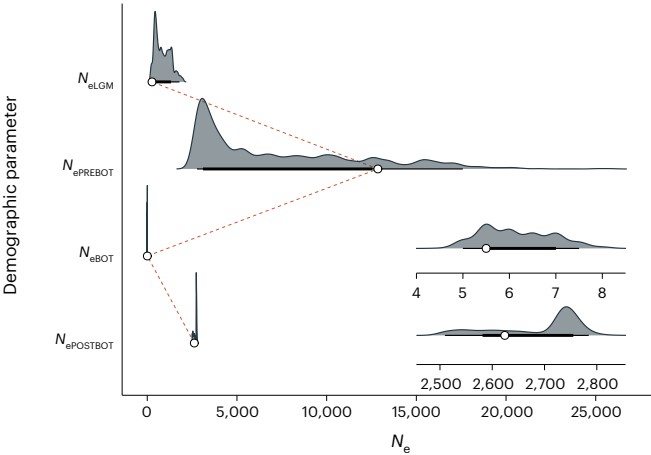

**Fig. 2 | Reconstruction of the recent demographic history of the northern elephant seal based on RAD sequencing data from 96 individuals.** The best-supported demographic model included post-glacial expansion followed by a recent bottleneck lasting for six generations and subsequent demographic recovery (see Extended Data Fig. 3 for details of the model and Extended Data Fig. 5 for the results of a sensitivity analysis). Shown are point estimates from the model with the best likelihood among 100 independent runs for each model (white points) and the distribution of estimates derived from 100 bootstrap replicates (grey shading represents the density distribution, thin lines are 95% CIs and thick lines are 66% CIs). $N_{eLGM}$ represents the effective population size during the last glacial maximum (LGM); $N_{ePREBOT}$ is the effective population size before sealing; $N_{eBOT}$ is the effective population size during the bottleneck; and $N_{ePOSTBOT}$ is the effective population size in the present day. The inserts in the bottom right of the figure represent the magnified distributions of $N_{eBOT}$ and $N_{ePOSTBOT}$.

are known to have been taken[25,26]. The third null model was otherwise identical but did not include a recent bottleneck (see Methods for details). All of the models started with an ancestral population at the end of the Last Glacial Maximum (LGM) that subsequently expanded until reaching the pre-bottleneck population size. The model with a bottleneck lasting for six generations received the highest support (Supplementary Table 4) and the simulated and observed SFSs were similar (Extended Data Fig. 4). Based on this model, $N_e$ during the LGM ($N_{eLGM}$), before sealing ($N_{ePREBOT}$), during the bottleneck ($N_{eBOT}$) and in the present day ($N_{ePOSTBOT}$) were estimated as 267 (95% CI = 227–1,722), 12,856 (2,828–20,275), six (95% CI = 5.0–7.5) and 2,624 (95% CI = 2,506–2,773), respectively (Fig. 2). Repeating this analysis using whole-genome sequencing (WGS) data from 20 individuals also revealed support for the six-generation bottleneck model, with a high degree of concordance between the parameter estimates (Supplementary Results and Discussion, Extended Data Fig. 5 and Supplementary Table 4). The low $N_{eBOT}$ estimate is consistent with the results of previous simulation studies based on mitochondrial DNA and microsatellites[56,57], as well as with Bartholomew and Hubb's[27] estimate of the population in 1890 numbering 20–100 individuals, assuming a census size ($N_c$)-to-$N_e$ ratio of approximately 10:1.

## Genetic load simulations

To investigate how the extreme demographic bottleneck described above may have shaped the genetic load of the northern elephant seal, we implemented forward genetic Wright–Fisher simulations using SLiM3 (ref. 58), as described in the Methods. The Wright–Fisher simulation model provides a flexible and generalizable modelling framework that is particularly well suited for characterizing variation in allele frequencies in response to demographic changes because the size of the simulated population can be explicitly controlled[59]. In these simulations, fitness is relative, with the genetic load affecting the probability of an individual being chosen as a reproducer but not affecting survival;

thus, population size can be held constant to a user-specified value. We simulated the demographic history of the northern elephant seal using point $N_e$ estimates from the best-supported demographic model based on the RAD sequencing data, as described in the Methods.

Figure 3 shows changes in various components of the genetic load of the simulated population from the generation before the start of the bottleneck until the present day. The total genetic load—quantified as the sum of the effect sizes of all deleterious mutations multiplied by their allele frequencies—shows a sigmoidal pattern, decreasing steeply between five and 15 generations after the start of the bottleneck and then flattening out (Fig. 3a). The initially slow decrease appears to reflect the fact that although large numbers of mutations are immediately lost from the population, at least partly due to strong genetic drift during the bottleneck, most of them have allele frequencies below 0.05 and therefore contribute relatively little to the total load of the population (Extended Data Fig. 6a,c). The surviving mutations drift to higher frequencies during the bottleneck and are subsequently purged or lost to genetic drift at different timepoints depending on their frequencies, with rarer mutations being lost earlier, until around 15 generations after the start of the bottleneck, by which time most of the purging and loss of deleterious alleles through drift has occurred (Extended Data Fig. 6b,c). The overall pattern does not appear to be influenced by changes in the effect size distribution of the surviving mutations (Extended Data Fig. 6d–f).

The realized load (that is, the fraction of the total load that is expressed) increases sharply after the start of the bottleneck, peaks around seven generations later and then gradually declines until the present day (Fig. 3b). This pattern is again due to surviving deleterious mutations drifting to higher frequencies and then being purged (Extended Data Fig. 6a–c). However, the realized load at the end of the simulation is around twice that of the pre-bottleneck population (Fig. 3b). The inbreeding load (that is, the fraction of the total load that is masked in the heterozygous state) shows a steep, continuous decrease from the start of the bottleneck until around 10–15 generations afterwards (Fig. 3c). This is due to a combination of the inbreeding load becoming increasingly expressed in homozygous genotypes (that is, being converted into the realized load), the loss of many low-frequency deleterious alleles via genetic drift, and purging by natural selection. In contrast, the drift load (that is, the decrease in fitness due to the continuous fixation of deleterious alleles) remains low during the bottleneck but increases in the recovering population before reaching an asymptote around ten generations after the start of the bottleneck (Fig. 3d). The delayed increase in the drift load after the onset of the bottleneck appears to be because it took several generations for previously rare deleterious alleles to drift all the way to fixation (Extended Data Fig. 6). These results are entirely consistent with theoretical predictions for the evolution of fitness in bottlenecked populations[20].

We investigated the robustness of our results by repeating the simulations described above while relaxing our assumptions, as well as using different datasets. Similar results were obtained (Supplementary Results and Discussion and Extended Data Fig. 7), suggesting that our inferences based on the Wright–Fisher models are reasonably robust to the underlying assumptions and datasets used for modelling. As the inbreeding load determines the strength of inbreeding depression[60–62], our results imply that inbreeding depression should be weaker in the post-bottleneck population of northern elephant seals compared with the pre-bottleneck population. Although we do not have an empirical historical baseline against which to compare our results, the fact that we do not find inbreeding depression for several key fitness components in the contemporary population aligns with this expectation. However, our results do not allow us to exclude the possibility that other fitness components unrelated to body condition or parasite and disease susceptibility might show inbreeding depression, especially where the underlying genetic architecture differs[11]. Indeed, Hoelzel

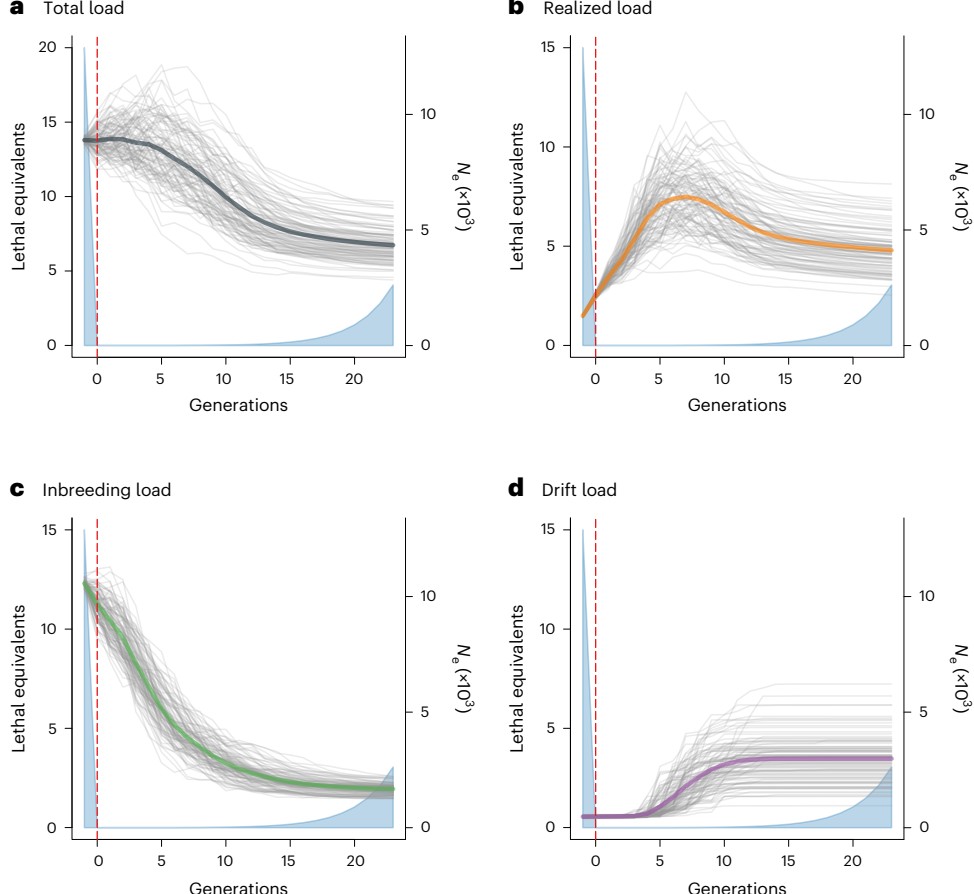

**Fig. 3 | Genetic load components of the simulated northern elephant seal population over time (measured in generations), starting from one generation before the bottleneck until the present day. a**, Total load, which corresponds to the total amount of lethal equivalents present in the population. **b**, Realized load, which represents the fraction of the total load that is expressed and which decreases the fitness of the population. **c**, Inbreeding load, which represents the fraction of the total load that is masked in the heterozygous state and which determines the strength of inbreeding depression in the presence of inbreeding. **d**, Drift load, which represents the subset of the realized load comprising deleterious mutations that have drifted to fixation. Thick coloured lines represent the averages of 100 forward genetic Wright–Fisher simulations (grey lines) with $N_e$ following the estimates from the best-supported demographic model derived from the RAD sequencing data (light blue shaded areas), as described in the Methods. The vertical dashed red lines indicate the onset of the bottleneck.

et al.[23] recently found evidence for inbreeding depression for lifetime reproductive success in female northern elephant seals. Reproductive success will be affected in complex ways by multiple factors, including the ones we have measured, and is therefore likely to be influenced by a larger number of genes across the genome. As our simulations show that the severe bottleneck will have dramatically decreased but not completely purged deleterious alleles, it is not unexpected that inbreeding depression can be detected for highly polygenic traits, such as reproductive success, but not for disease traits, many of which are likely to be oligogenic[63] and should therefore be purged more efficiently. Hence, the differences between these two studies emphasize the dependence of the outcomes of bottlenecks not only on the severity of demographic declines but also on the traits in question and their underlying genetic architectures.

**Genomic inbreeding and individual genomic mutation loads**

To characterize patterns of genetic diversity and estimate individual genomic mutation loads, we generated WGS data (median coverage = 18.6×) for 20 northern elephant seals. To provide a comparative perspective, we additionally generated WGS data (median coverage = 19.3×) for 20 southern elephant seals. The two sister species diverged only around 0.6–4.0 million years ago[64,65] but experienced markedly different recent demographic histories, with only

the northern elephant seal having been hunted to the brink of extinction[31]. We found that the number of segregating sites differed by over an order of magnitude between the species, with 1,234,849 SNPs being called in the northern elephant seal compared with 14,900,073 SNPs in the southern elephant seal. Nucleotide diversity ($\pi$) was nearly an order of magnitude lower in the northern elephant seal ($\pi = 0.0003$) compared with the southern elephant seal ($\pi = 0.0017$) and the variation in $\pi$ in 100 kilobase (kb) windows along the genome was also lower in the northern elephant seal (mean ± s.d. = 0.0004 ± 0.0009 versus 0.0016 ± 0.0023, respectively; Fig. 4a). Similarly, individual genome-wide heterozygosity was lower in the northern elephant seal (mean ± s.d. = 0.00018 ± 0.000005 versus 0.00149 ± 0.000034; Fig. 4b). These results are consistent with previous empirical estimates of very low genetic diversity in the northern elephant seal[56,66,67] and are indicative of a very low harmonic mean $N_e$ over the 23 generations since the onset of the bottleneck (Supplementary Results and Discussion).

To infer genomic inbreeding, we quantified the proportion of each individual's genome in runs of homozygosity ($F_{ROH}$) using a conservative minimum ROH length threshold of 1 megabase (Mb). Consistent with the above results, $F_{ROH}$ was substantially higher in the northern elephant seal (mean ± s.d. = 0.21 ± 0.03) compared with the southern elephant seal (mean ± s.d. = 0.03 ± 0.02; Fig. 4c). However, these values may underestimate the true magnitude of inbreeding as decreasing the minimum

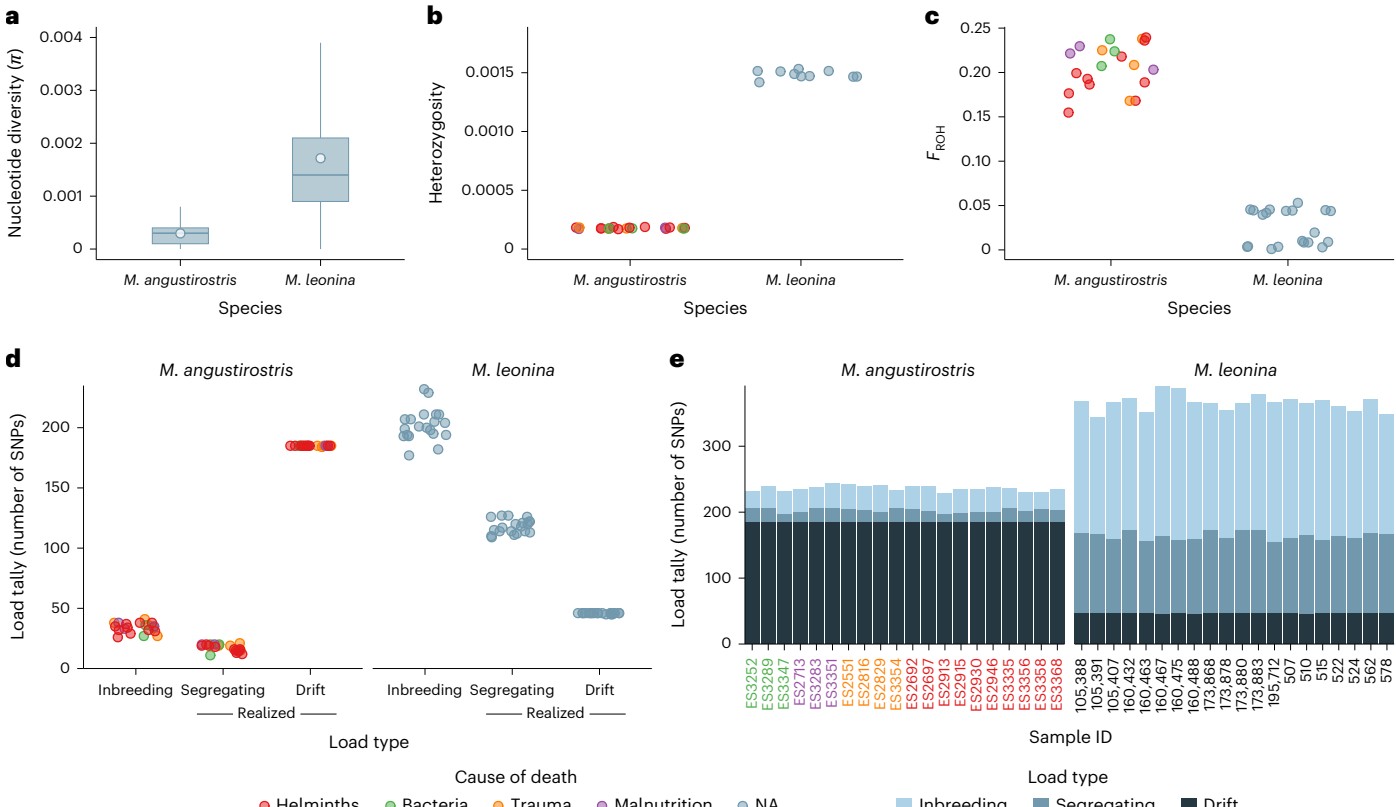

Cause of death ● Helminths ● Bacteria ● Trauma ● Malnutrition ● NA

Load type ▪ Inbreeding ▪ Segregating ▪ Drift

**Fig. 4 | Estimates of genetic diversity, inbreeding and genomic mutation loads based on whole-genome resequencing data from northern (*M. angustirostris*) and southern (*M. leonina*) elephant seals (*n* = 20 each). a**, Box-and-whiskers plots showing the distribution of nucleotide diversity ($\pi$) within non-overlapping 1 Mb windows along the genome. The white points show genome-wide mean $\pi$ values, the boxes indicate the first and third quartiles, the horizontal lines indicate the medians and the whiskers extend up to 1.5× the interquartile range. Outliers have been omitted for clarity. **b**, Mean individual genome-wide heterozygosity. **c**, The magnitude of individual inbreeding as expressed by the genomic inbreeding coefficient $F_{ROH}$ (with a minimum ROH length threshold of 1 Mb). Smaller ROH length thresholds resulted in larger

$F_{ROH}$ values, as shown in Extended Data Fig. 8, but the overall pattern remained unchanged. **d**, Individual genomic mutation loads based on tallies of putatively deleterious SNPs detected in each individual. These were classified into the inbreeding load (heterozygous SNPs) and the realized load (homozygous SNPs). The realized load was further decomposed into the segregating load (mutations that are variable within a species) and the drift load (mutations that are fixed within a species). **e**, Individual genomic mutation loads decomposed into the inbreeding, segregating and drift loads, as shown in the legend. In **b**–**e**, the northern elephant seal individuals are colour coded according to their most likely cause of death, as shown in the legend. NA refers to the southern elephant seal individuals, which were not assigned to fitness categories.

ROH length threshold from 1 Mb to 1 kb increased the mean ± s.d. $F_{ROH}$ to 0.67 ± 0.01 and 0.15 ± 0.02, respectively (Extended Data Fig. 8). Nevertheless, the strong relative difference in inbreeding between the two species remained regardless of the ROH length threshold.

Genetic loads can be quantified both at the population and individual levels[12]. We therefore estimated individual genomic mutation loads from the WGS data, using SNPeff[68] to identify derived alleles at variable sites predicted to disrupt protein function (specifically, high-impact and loss-of-function variants). It is important to note that this approach quantifies the number of putatively deleterious mutations within each individual genome but is not informative about their selection (*s*) or dominance (*h*) coefficients. Therefore, in contrast with the forward genetic simulations, where mutation loads are expressed as lethal equivalents at the population level, our genomic mutation load estimates represent rough proxies corresponding to tallies of predicted deleterious mutations at the individual level. We calculated the total number of derived deleterious mutations as a proxy measure of the total load, the number of derived alleles in the heterozygous state as a proxy measure of the inbreeding load and the number of derived alleles in the homozygous state as a proxy measure of the realized load. We then decomposed the realized load into mutations that are variable within the focal species (hereafter referred to as the segregating load[21]) and mutations that are fixed within the focal species (here assumed

to represent the drift load). Fixed mutations in the focal species were estimated by subsetting sites that segregated between the two species to include only those sites that were invariant in the focal species. Mutations contributing to the inbreeding, segregating and drift loads were broadly distributed across the genomes of both species (Supplementary Results and Discussion and Extended Data Fig. 9).

Marked differences were found in both the magnitude and composition of the genomic mutation loads of the two species (Fig. 4d,e). In line with the results of the Wright–Fisher simulations, the total load was substantially lower in the northern elephant seal (235.9 ± 4.1 s.d. deleterious alleles per individual) than in the southern elephant seal (365.9 ± 12.2 s.d. deleterious alleles per individual). Similarly, the inbreeding load was lower in the northern elephant seal (33.9 ± 4.2 s.d. versus 202.2 ± 13.2 s.d. heterozygous deleterious alleles per individual). The realized load showed the opposite pattern, being higher in the northern elephant seal (202.1 ± 3.0 s.d. versus 163.8 ± 5.9 s.d. homozygous deleterious alleles per individual), mainly due to the drift load being higher (184.9 ± 0.2 s.d. versus 45.9 ± 0.4 s.d. homozygous deleterious alleles per individual). However, our sample sizes do not allow us to distinguish between mutations that have become fixed and mutations that have drifted to very high frequency (>0.975), potentially resulting in the overestimation of mutations contributing to the drift load. Additionally, our approach cannot discriminate between

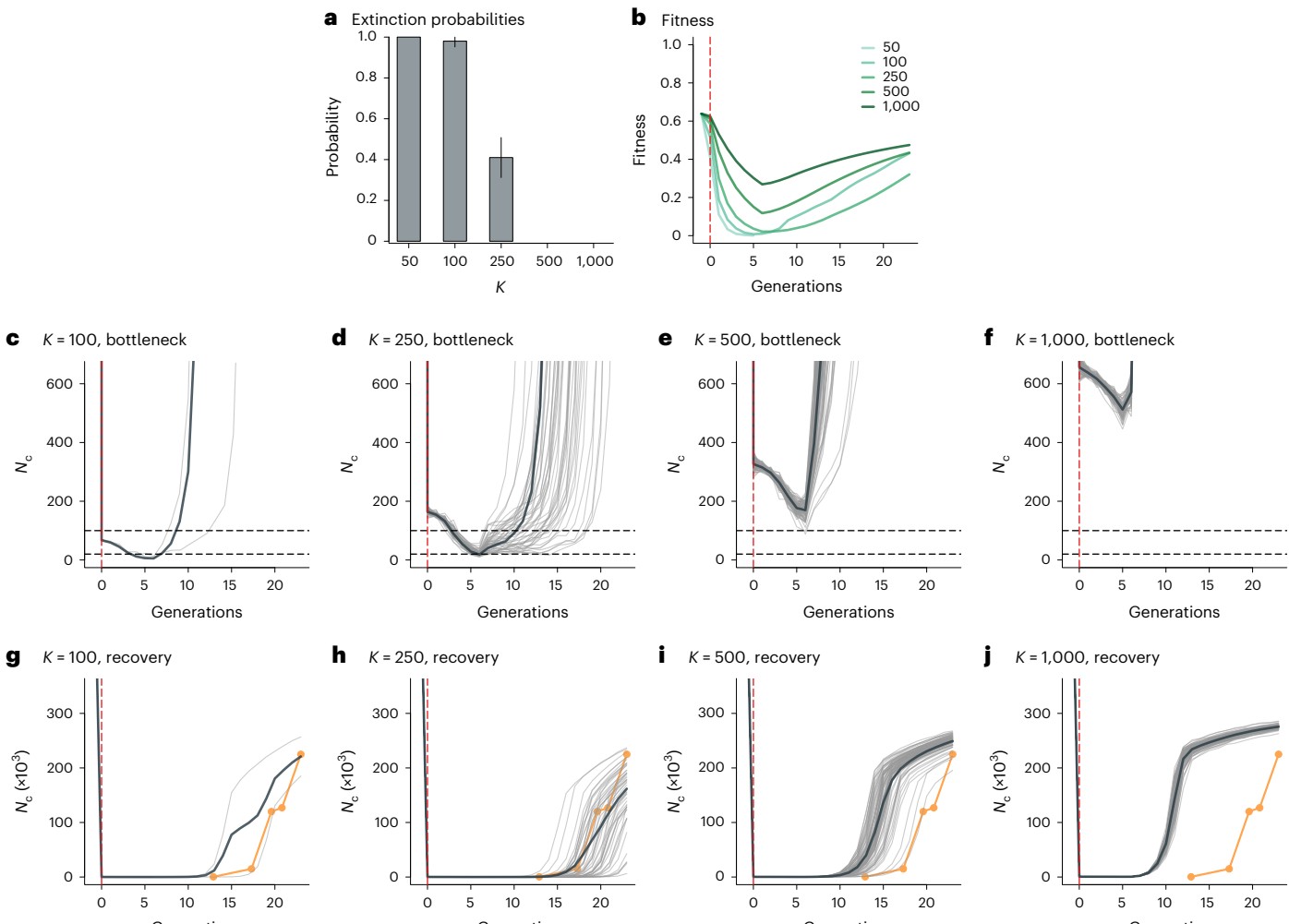

**Fig. 5 | Extinction probabilities and fitness and census population size trajectories of the simulated northern elephant seal population over time (shown in generations), starting from one generation before the bottleneck until the present day.** Results are shown for different sets of 100 non-Wright–Fisher simulations that varied in the carrying capacity ($K$) of the population during the bottleneck. **a**, Extinction probabilities. **b**, Population fitness averaged over the 100 simulations separately for each set of simulations, as shown in the legend. **c**–**f**, Census population sizes ($N_c$) of the surviving simulated populations for $K = 100$ (**c**), $K = 250$ (**d**), $K = 500$ (**e**) and $K = 1,000$ (**f**) during the bottleneck. The horizontal dashed lines indicate the estimated minimum $N_c$ of the northern elephant seal population (range = 20–100 individuals) according to Bartholomew and Hubbs[27]. **g**–**j**, Population growth for $K = 100$ (**g**), $K = 250$ (**h**), $K = 500$ (**i**) and $K = 1,000$ (**j**). The thick dark grey lines represent average $N_c$ across the 100 simulations (light grey lines). The orange points (joined by orange lines) indicate empirical $N_c$ estimates from Bartholomew and Hubbs[27], Le Boeuf and Bonnell[28], Stewart et al.[29] and Lowry et al.[30]. The vertical dashed red lines indicate the onset of the bottleneck.

deleterious mutations that reached fixation during the bottleneck and those that fixed divergently between the two elephant seal species as part of the speciation process. Nevertheless, the lower average inbreeding load of the northern elephant seal, together with the fact that most individuals appear to carry similar numbers of deleterious alleles regardless of their disease status (Fig. 4d,e), is again consistent with the lack of detectable inbreeding depression in this study.

### Extinction probability, fitness and population recovery

To investigate how close the northern elephant seal came to extinction and to evaluate the potential effects of the bottleneck on population recovery, we implemented non-Wright–Fisher simulations with SLiM3, where deleterious mutations impact the absolute fitness of individuals by affecting annual survival probabilities. Non-Wright–Fisher simulations are ideally suited for this purpose because they rely on the absolute fitness of simulated individuals combined with a user-specified carrying capacity ($K$) to dynamically determine $N_c$, which can go to zero when fitness is particularly low[58]. Furthermore, demographic

stochasticity associated with genetic and life history traits such as the reproductive system and age-specific mortality can be incorporated. We modelled a polygynous mating system with age-specific reproduction and mortality according to published estimates of northern elephant seal life history traits (see Methods for details). To investigate the range of possible outcomes for bottlenecks of varying intensity, we ran 100 simulations for each of five scenarios that differed in the carrying capacity of the population during the bottleneck, from $K = 50$–1,000.

Figure 5 summarizes the extinction probabilities (Fig. 5a) and fitness (Fig. 5b) and demographic trajectories (Fig. 5c–j) of the simulated populations. We found that the probability of extinction was strongly dependent on bottleneck strength. All of the simulated populations with $K = 50$ went extinct, suggesting that it is unlikely that the northern elephant seal experienced such an extreme bottleneck. The extinction probability was also high at 98% ($\pm$2.8 s.d.) for $K = 100$, whereas it decreased to 41% ($\pm$9.8 s.d.) for $K = 250$ and to zero for $K \geq 500$ (Fig. 5a). Based on their interpretation of seal counts from the early 1900s, Bartholomew and Hubbs[27] estimated that the population decreased to

below 100 individuals and possibly as few as 20 individuals. In line with this, the two surviving simulated populations for the $K = 100$ scenario decreased to seven and 12 individuals, respectively (Fig. 5c), whereas the surviving simulated populations for the $K = 250$ scenario decreased to an average $N_c$ of 22.7 ± 9 s.d. (Fig. 5d), making these arguably the most realistic scenarios out of those we tested. In contrast, the simulated populations for $K = 500$ and $K = 1,000$ decreased to an average $N_c$ of 164.0 ± 23.4 s.d. (Fig. 5e) and 510.7 ± 21.0 s.d. (Fig. 5f), respectively.

Next, we investigated the impact of bottleneck intensity on the timing of population recovery. $N_c$ estimates based on empirical data[27–30] indicate that the northern elephant seal population comprised just a few thousand individuals during the first few decades of the early twentieth century and rapidly increased starting from the 1960s (orange lines in Fig. 5g–j). Simulated population trajectories for the surviving populations with $K = 100$ and $K = 250$ closely matched this empirical trajectory (Fig. 5g,h). Larger values of $K$ resulted in progressively earlier population recoveries, with the contemporary $N_c$ being reached around a century earlier in the population with the weakest simulated bottleneck ($K = 1,000$; Fig. 5j). These results are consistent with the different fitness trajectories emerging from the simulations with varying bottleneck intensities, where fitness is a function of the realized load (Fig. 5b).

Finally, we aimed to rule out the possibility that the demographic patterns emerging from the non-Wright–Fisher simulations could be due to demographic stochasticity associated with bottlenecks of different strength and have little to do with deleterious genetic variation. We therefore implemented a series of neutral models that were identical in all respects to the previous non-Wright–Fisher simulations, but which did not simulate any deleterious mutations. Three striking differences were observed (Extended Data Fig. 10). First, none of the simulated populations without deleterious mutations went extinct. Second, they showed earlier demographic recoveries than simulations that included deleterious mutations. Third, they rapidly attained the maximum final $N_c$ determined by the carrying capacity of the simulations. This reinforces the notion that differences in the genetic load of populations experiencing bottlenecks of different strength can play an important role in determining their rate of recovery and suggests that deleterious genetic variation strongly impacted the post-bottleneck population dynamics of the northern elephant seal.

## Conclusions

The northern elephant seal is a classic example of a wild vertebrate that is believed to have purged at least part of its genetic load as a result of a severe bottleneck[7]. However, there is little empirical evidence for this assertion other than the widespread perception of the "remarkable demographic vitality"[29] of the recovering population. More generally, it is challenging to demonstrate purging in wild populations because inbreeding depression usually cannot be measured before and after purging has taken place[11]. We circumvented this issue by ground-truthing our empirical results concerning the contemporary level of inbreeding depression with simulations and individual genomic mutation load estimates based on WGS data. This produced three complementary lines of evidence in support of the argument that the bottleneck resulted in a decrease in the inbreeding load of the northern elephant seal: (1) the absence of detectable inbreeding depression for body mass, blubber thickness and susceptibility to parasites and disease within the contemporary population, despite the presence of a clear signal of inbreeding; (2) simulations indicating a substantial decrease in the total and inbreeding load of the northern elephant seal population due to the bottleneck; and (3) genomic evidence of fewer derived deleterious mutations in the northern versus southern elephant seal. The extent of purging in other natural systems remains uncertain[11], but laboratory studies indicate that the demographic impact of purging tends to be associated with a high risk of extinction[69]. This is consistent with the results of our non-Wright–Fisher simulations, which revealed an appreciable risk of extinction (ranging from 98% ± 2.8 s.d. for $K = 100$

to 41% ± 9.8 s.d. for $K = 250$) for those scenarios most closely resembling the known recent demographic history of the northern elephant seal, emphasizing that the persistence of this species was a fortunate outcome. Part of the explanation for why the northern elephant seal persisted may be that selection associated with inbreeding depression mainly impacted traits that affect relative but not absolute fitness (that is, selection was mainly soft[70]).

Also in accordance with theoretical predictions[8,12,20], our simulations revealed a marked increase in the realized genetic load of the northern elephant seal population as a result of the bottleneck. This increase in the realized load appears to have slowed down the demographic recovery of the population. Although many deleterious alleles were almost certainly purged during and shortly after the bottleneck, strong genetic drift after the population crash means that many other previously rare deleterious alleles are expected to have drifted to high frequency or fixation, resulting in a net increase in the genetic load. Our results, combined with those of Hoelzel et al.[23], who reported inbreeding depression for female lifetime reproductive success in northern elephant seals, imply that the genomic consequences of the severe anthropogenic bottleneck hindered what is considered by many to be "one of the most remarkable population recoveries of any mammal"[31].

To conclude, our findings emphasize the complexity of population-level responses to demographic declines and their reliance on demographic history. In the case of the northern elephant seal, a severe population bottleneck resulted in a net decrease in fitness and population growth. Furthermore, whole-genome resequencing data from the two elephant seal sister species suggest that the bottleneck was associated with a substantial decrease in genome-wide diversity, in line with previous studies based on smaller numbers of genetic markers[57,65–67]. This loss of genetic diversity may constrain the ability of this species to adapt to future challenges[22,71]. Given that anthropogenic pressures are causing species declines on an unprecedented scale, there is an urgent need to better understand the interplay between demographic histories, genetic diversity and fitness, and how these dynamics unfold across species.

## Methods

### Northern elephant seals

Tissue samples were obtained from juvenile elephant seals that died from various causes between 2006 and 2012 at The Marine Mammal Center (Sausalito, CA, USA) following stranding along the California coast from San Luis Obispo to Humboldt County. At admittance, the animals were weighed (kg) and their blubber thickness (cm) was measured over the sternum. Within 24 h of death, each animal was subjected to a standardized postmortem examination. Helminth infections were identified by examining the respiratory tract, heart and great blood vessels and gastrointestinal tract. Representative samples of internal organs were fixed in 10% neutral buffered formalin, embedded in paraffin, sectioned at 5 μm and stained with haematoxylin and eosin for light microscopy to examine the tissues histologically. As described by Colegrove et al.[72] the cause of death was defined as the disease or condition that was the most likely cause of the animal's death based on all of the available information. Each individual was assigned to one of the following categories:

(1) Helminth infection ($n = 62$): clinical manifestations of *Otostrongylus circumlitus* infection, including neutrophilia, disseminated intravascular coagulation and fatal inflammatory reaction.
(2) Bacterial infection ($n = 28$): primary infection with a specific microbial organism (for example, *Leptospira* or *Salmonella* species) or generalized non-specific bacterial infection following trauma or malnutrition.
(3) Protozoan infection ($n = 4$): infection with a protozoal parasite (for example, *Toxoplasma gondii*, *Sarcocystis* species or *Eimeria* species).

 

(4) Trauma ($n = 19$): wounds with no complicating generalized infections, including boat strikes, bite wounds, lacerations and musculoskeletal injuries.

(5) Malnutrition ($n = 88$): poor body condition with no other obvious infections or lesions.

(6) Congenital defects ($n = 18$): the presence of a major congenital defect that contributed to poor health, either causing death or resulting in euthanasia due to a poor veterinary prognosis.

Finally, a ~5 mm³ skin sample was collected from the foreflipper of each animal with a scalpel and stored in 95% ethanol at −20 °C for subsequent analysis. All animal care and sampling procedures were authorized by the National Marine Fisheries Service (MMPA permit number 18786) and approved by The Marine Mammal Center's internal animal care and use committee.

## Southern elephant seals

Tissue samples were obtained from 20 southern elephant seals (one adult female, three adult males and 16 pups) from a breeding colony at Half Moon Beach, Cape Shirreff in the South Shetland Islands (62° 28′ 37.4′′ S, 60° 46′ 45.0′′ W) between 2008 and 2015 (see Nichols et al.[73] for details). Adults were sampled from the flanks using a 2 mm sterile disposable Miltex biopsy punch (Thermo Fisher Scientific). Pup skin samples were collected from a rear flipper using a tag hole punch or a 2 mm sterile disposable Miltex biopsy punch. The samples were immediately transferred to 95% ethanol and stored at −20 °C until subsequent analysis. All sampling was conducted in accordance with Marine Mammal Protection Act permit numbers 16472-01 and 774-1847-04, granted by the Office of Protected Resources, National Marine Fisheries Service (Antarctic Conservation Act permit numbers 2012-005 and 2008-008). The protocols used in this study were also reviewed and approved by the US National Marine Fisheries Service Southwest and Pacific Islands regions' Institutional Animal Use and Care Committee (approval documents SWPI2011-02 and SWPI2014-03).

## Microsatellite genotyping

Total genomic DNA was extracted from the northern elephant seal samples using a modified phenol–chloroform protocol and genotyped at 22 polymorphic microsatellite loci (Supplementary Table 1). The loci were PCR amplified in four separate multiplexed reactions using a Type-it kit (Qiagen). The following PCR profile was used: initial denaturation of 5 min at 94 °C; 28 cycles of 30 s at 94 °C, 90 s at the annealing temperature ($T_a$ °C) specified for each multiplex reaction in Supplementary Table 1, and 30 s at 72 °C, followed by a final extension of 30 min at 60 °C. Fluorescently labelled PCR products were then resolved by electrophoresis on an ABI 3730xl capillary sequencer and allele sizes were scored using GeneMarker version 1.95. To ensure high genotype quality, all traces were manually inspected and any incorrect calls were adjusted accordingly. We also quantified the genotyping error rate for the resulting dataset by independently repeat genotyping 85 samples between one and five times each. This was very low at 0.0016 per locus or 0.0013 per allele. Finally, deviations from Hardy–Weinberg equilibrium were calculated for each microsatellite locus using chi-squared tests and exact tests based on 1,000 Monte Carlo permutations of the dataset[74] implemented in the R package pegas 1.0-1 (ref. [75]). The resulting $P$ values were corrected for the false discovery rate using the R package qvalue[76]. The R package adegenet[77] was also used to calculate the observed and expected heterozygosity of each locus.

## RAD sequencing

A representative subset of 96 northern elephant seals (33 animals with helminth infection, 19 with bacterial infection, two with protozoan infection, 11 with trauma, 22 with malnutrition and nine with congenital defects) was chosen for RAD sequencing. The libraries were prepared using a modified protocol from Etter et al.[78] with minor modifications

as described by Hoffman et al.[37]. Briefly, 400 ng genomic DNA from each individual was separately digested with SbfI followed by the ligation of P1 adaptors with a unique 6 bp barcode for each individual in a RAD sequencing library, allowing the pooling of 16 individuals per library. Libraries were sheared with a Covaris S220 and agarose gel size selected to 300–700 bp. Following 15–17 cycles of PCR amplification, the libraries were further pooled using eight different i5 indices before 250 bp paired-end sequencing on two Illumina HiSeq 1500 lanes. This resulted in a total of 315,558,810 paired-end 250 bp Illumina sequence reads.

The quality of the raw sequences was checked using FastQC[79] and low-quality reads were trimmed using the fastx_trimmer module within the FASTX-Toolkit (http://hannonlab.cshl.edu/fastx_toolkit). Subsequently, the reads were demultiplexed using the process_radtags module within the Stacks pipeline[80] and any reads containing uncalled bases or reads with an average phred scaled quality score below ten within sliding windows comprising 15% of the length of the read were discarded. We then used the bwa-mem algorithm in BWA[81] with default parameters to align the demultiplexed reads to the *M. angustirostris* reference genome, available via the National Center for Biotechnology Information (NCBI; RefSeq identification code GCF_021288785.2). The resulting SAM files were converted to BAM format and sorted by coordinate using SAMtools[82]. Next, Picard tools (https://broadinstitute.github.io/picard) was used to add read groups to individual BAM files and to mark and remove PCR duplicates. Finally, variant discovery and calling was implemented using the HaplotypeCaller module within GATK[83] and the resulting 321,124 raw genotypes were filtered using VCFtools[84] and PLINK[85] to retain only high-quality biallelic SNPs. Specific filtering steps included: (1) retaining only individual genotypes with a depth of coverage and genotype quality of greater than five; (2) removing loci that could not be called in more than 50% of individuals; (3) removing loci whose coverage exceeded twice the mean coverage of the dataset (19.06×); (4) removing loci that deviated significantly from Hardy–Weinberg equilibrium based on a $P$ value threshold of 0.01 after having implemented mid-$P$ adjustment, as described by Graffelman and Moreno[86]; (5) removing loci with a minor allele frequency below 0.01; and (6) removing samples with more than 50% of missing data. The final quality-filtered dataset comprised 74 individuals genotyped at 15,051 SNPs.

## Population structure

To test for the presence of population genetic structure, we subjected both the microsatellite and RAD sequencing datasets to PCA using the R package adegenet[77,87]. Specifically, we applied the function dudi.pca to the microsatellite data and the glPca function to the SNP dataset. The glPca function is specifically designed to efficiently compute PCA on large SNP datasets.

## Identity disequilibrium

To quantify the variance in inbreeding, we calculated the two-locus heterozygosity disequilibrium ($g_2$)[35] for the microsatellite and RAD sequencing datasets using the R package inbreedR[88] version 0.3.3. The 95% CIs of $g_2$ were determined based on 100 permutations of the dataset and bootstrapping 10,000 times over individuals. InbreedR was also used to quantify individual sMLH for both the microsatellite and RAD sequencing datasets.

## Inbreeding depression

To test for inbreeding depression for body mass and blubber thickness, we constructed Bayesian linear mixed models using the R package brms[89] version 2.19.0. The predictor variable was $z$-transformed sMLH, and sex and the month and year of admittance were included as random effects, as follows:

$$\text{mass}_{ij} = \beta_0 + \beta_1 \times \text{sMLH}_{ij} + b_i + u_j + \upsilon_k$$

$$\text{blubber}_{ij} = \beta_0 + \beta_1 \times \text{sMLH}_{ij} + b_i + u_j + \upsilon_k$$

where:

mass$_{ij}$ represents the observed value of the response variable mass for observation $i$ within the levels of the grouping variables sex ($i$), month ($j$) and year ($k$);

blubber$_{ij}$ represents the observed value of the response variable blubber thickness for observation $i$ within the levels of the grouping variables sex ($i$), month ($j$) and year ($k$);

$\beta_0$ and $\beta_1$ are the fixed-effects coefficients for the intercept and the predictor variable sMLH, respectively;

$b_i$ represents the random intercept for each level of the grouping variable sex;

$u_j$ represents the random intercept for each level of the grouping variable month; and

$v_k$ represents the random intercept for each level of the grouping variable year.

To investigate whether inbreeding is related to the most likely cause of death, we constructed Bayesian GLMMs separately for each of the six categories (helminth infection, bacterial infection, protozoan infection, trauma, malnutrition and congenital defects). For this, we used binomial classifications (that is, animals whose most likely cause of death was helminth infection were classified as 1 for helminth infection and 0 for all of the other categories). Again, $z$-transformed sMLH was fitted as the predictor variable, and sex and the month and year of admittance were included as random effects:

$$\log\left[\frac{p_{ij}}{1-p_{ij}}\right] = \beta_0 + \beta_1 \times \text{sMLH}_{ij} + b_i + u_j + v_k$$

where:

$p_{ij}$ represents the probability that the binary response variable category (for example, helminth infection and so on) is equal to 1 (died of the respective disease or condition) for observation $i$ within the levels of the grouping variables sex (indexed by $i$), month (indexed by $j$) and year (indexed by $k$).

We also ran a similar analysis, but instead of using the binary classification of the most likely cause of death, we constructed a single multinomial GLMM with a categorical response variable, where each category was compared with the reference category trauma:

$$\text{logit}\left[P(Y_{ij} \leq m)\right] = \alpha_m - \beta \times \text{sMLH}_{ij} + b_i + u_j + v_k \text{ for } m = 1, 2, \ldots k-1$$

where:

$\text{logit}\left[P(Y_{ij} \leq m)\right]$ is the log-odds of $Y_{ij}$ being less than or equal to category $m$; and

$\alpha_m$ is the threshold parameter associated with category $m$.

Lastly, to investigate whether inbreeding differed between animals that died from trauma and those that died from other causes, we constructed Bayesian GLMMs using a binomial classification where animals whose most likely cause of death was trauma were classified as 0 and all other categories apart from trauma (that is, helminth infection, bacterial infection, protozoan infection, malnutrition and congenital defects) were classified as 1. Again, $z$-transformed sMLH was fitted as the predictor variable, and sex and the month and year of admittance were included as random effects:

$$\log\left[\frac{p_{ij}}{1-p_{ij}}\right] = \beta_0 + \beta_1 \times \text{sMLH}_{ij} + b_i + u_j + v_k$$

where:

$p_{ij}$ is the probability that the binary response variable category is equal to 1 (did not die from trauma) for observation $i$ within the levels of the grouping variables sex (indexed by $i$), month (indexed by $j$) and year (indexed by $k$).

All of the above models were implemented separately for the microsatellite and RAD sequencing datasets. Three independent

Markov chains were run for 100,000 iterations, using a thinning interval of 100 and burn-in of 50,000 iterations. We used generic weakly informative priors for the population-level effects. Model diagnostics, including autocorrelation and R hat statistics, and effective sampling sizes were generated using the R package bayesplot version 1.10.0 (ref. 90). All statistical analyses were implemented in R version 3.6.3 (ref. 91) with Rstudio version 1.3.1093 using the tidyverse R package[92] version 1.3.1.

### Demographic reconstruction

To reconstruct past changes in the effective population size of the northern elephant seal, we performed demographic inference based on the folded SFS derived from genotype likelihoods. Specifically, the program ANGSD[93] was used to calculate genotype likelihoods based on the BAM files obtained after mapping the sequencing reads to the reference genome and filtering to remove alignments to the sex chromosome. This was implemented while retaining only uniquely mapped reads with a minimum mapping quality of 20, for which we also calculated base alignment quality scores to reduce errors deriving from misalignments around indels[94]. Moreover, we retained only sites present in all individuals that had a minimum and maximum depth of coverage of 288 and 1,600, respectively, across all individuals. The resulting genotype likelihoods were then used as input to the ANGSD module realSFS to estimate the empirical folded SFS.

Demographic inference was implemented using the coalescent simulator fastsimcoal2 (ref. 95), which we used to compare three alternative demographic models (Extended Data Fig. 3). The first model included a recent bottleneck spanning the peak of commercial exploitation of the northern elephant seal in the nineteenth century. The bottleneck was fixed between 23 and 17 generations ago, corresponding to the known period of intensive harvesting (1810–1860)[24,25] assuming a generation time of 8.7 years[96]. The second model accounted for the fact that northern elephant seals continued to be hunted at a lower level until the end of the nineteenth century[25,26]. Accordingly, the bottleneck was fixed between 23 and 13 generations ago. The third model did not include a recent bottleneck and therefore represented a null model. All of the models included a period of post-glacial population expansion, whose end, measured in generation ago, was inferred from the model. We denoted this time point (i.e. the end of the post-glacial expansion) as $T_{se}$. We included post-glacial expansion in our models because a previous comparative study of pinnipeds based on microsatellites found greater support in northern elephant seals for a demographic model that included a recent bottleneck and post-glacial expansion than a model that included only a recent bottleneck[67].

In all of these models, the defined initial search range for the current effective population size ($N_{e\text{POSTBOT}}$) was log uniformly distributed between 5,000 and 40,000. For the bottleneck models, the defined initial search range for the effective population size before sealing ($N_{e\text{PREBOT}}$) was log uniformly distributed between 5,000 and 40,000, whereas the defined initial search range for the effective population size during the bottleneck ($N_{e\text{BOT}}$) was uniformly distributed between one and 50. Then, we defined the initial search range for the effective population size after the LGM ($N_{e\text{LGM}}$) between 50 and 4,000 and set the LGM to 2,100 generations ago, corresponding to approximately 19,000 years ago[97]. Finally, the initial search range for $T_{se}$ was uniformly distributed between 100 and 1,000 generations ago. Note that the composite maximum likelihood approach implemented in fastsimcoal2 uses these search ranges solely as starting values and the resulting parameter estimates can therefore exceed the upper limits[55].

A total of 100 replicate runs were performed for each model, including 100 estimation loops with 100,000 coalescent simulations. We did not include singletons in the simulations, as these can be biased when the sequence coverage is low[98]. Out of the 100 replicates for each model, the run with the highest maximum likelihood was retained. The best model was then determined based on the delta likelihood values

(the difference between the estimated and observed likelihoods) and Akaike's information criterion values. Finally, we investigated the uncertainty of our parameter estimates by bootstrapping the data 100 times over individuals with replacement and using ANGSD to generate corresponding SFSs. For each of these 100 SFSs, the parameters were then re-estimated based on 100 replicate runs, each including 100 estimation loops with 100,000 coalescent simulations. For each SFS, the run with the top maximum likelihood was retained and used for the bootstrap distribution. For each parameter, 95% CIs were then calculated based on the resulting 100 bootstrap estimates.

Finally, to explore the sensitivity of our results to different types of input data, we repeated the demographic inference described above using SFSs obtained from WGS data from 20 northern elephant seals (for details, see below). The genotypes were filtered to include only autosomes using VCFtools and the corresponding SFSs were obtained using easySFS (https://github.com/isaacovercast/easySFS). Demographic inference was then implemented as described above for the RAD sequencing data, using the same models and priors.

### Wright–Fisher simulations

To investigate how the inferred bottleneck may have impacted the genetic load of the northern elephant seal, we implemented forward genetic simulations with the software SLiM3 (ref. [58]). Specifically, we ran 100 Wright–Fisher simulations where we modelled the demographic history of the species since the LGM using point $N_e$ estimates derived from the best-supported demographic model based on the RAD sequencing data (see 'Results'). Starting from one generation before the bottleneck, we then quantified the following components of the genetic load of each simulated generation according to Bertorelle et al.[12]:

(1) Total load. This represents the total number of lethal equivalents present in the population. It is quantified as the sum of the effect sizes of all deleterious mutations multiplied by their allele frequencies. It is thus independent of genotype frequencies and incorporates both the component of the genetic load that is expressed (that is, the realized load; see below) and the component of the genetic load that is not expressed (that is, the inbreeding load; see below).

(2) Realized load. This is the fraction of the total load that is expressed and which therefore actively decreases the fitness of the population. It is determined by homozygous deleterious mutations and heterozygous deleterious mutations that are not fully recessive, whose effects are scaled by their dominance coefficients. Therefore, the realized load is quantified as the sum of the effect sizes of all homozygous mutations multiplied by their genotype frequencies plus the sum of the effect sizes of heterozygous mutations multiplied by their genotype frequencies and respective dominance coefficients.

(3) Inbreeding load. This is the fraction of the total load that is masked in the heterozygous state. It is quantified by subtracting the realized load from the total load. This is the load component that determines inbreeding depression, as inbreeding unmasks the effects of deleterious mutations that are shielded from selection in the heterozygote state.

(4) Drift load. This is the subset of the realized load that is represented exclusively by deleterious mutations that have drifted to fixation. We calculated this as the sum of the effect sizes of all fixed deleterious mutations.

We simulated the entire northern elephant seal genome, comprising all 17 autosomes, and allowed only deleterious mutations to arise. A burn-in of $10 \times N_e$ generations was implemented to establish an equilibrium level of genetic diversity through mutation–selection balance. We modelled the evolution of deleterious mutations based on the available information to date[22]. First, these mutations were modelled to appear at a rate greater than one ($U \cong 1.2$), which is in accordance with empirical estimates from fruit flies[99] and humans[100]. Second, the distribution of fitness effects of the deleterious mutations was modelled as strongly bimodal, with the majority of mutations having small to moderate effects while a minority were lethal or semi-lethal[101]. This was achieved by sampling $|s|$ from a gamma distribution with the mean and shape parameter equal to −0.04 and 0.2, respectively. Third, dominance coefficients ($h$) were modelled so that nearly neutral mutations were slightly recessive and highly deleterious mutations were nearly fully recessive, in accordance with empirical observations in fruit flies[102] and yeast[103]. For this we assumed the relationship between $h$ and $|s|$ provided by Deng and Lynch[104]. Finally, deleterious mutations were allowed to appear throughout the genome—an assumption we believe to be realistic as deleterious mutations are known to arise not only within exons but also in regulatory elements and ultra-conserved genomic regions[21]. Nevertheless, for comparison, we also re-ran the simulations while allowing deleterious mutations to arise only within exons.

Finally, we incorporated the uncertainty associated with our demographic estimates by re-running the forward genetic simulations described above using the $N_e$ estimates obtained from the bootstrapped SFSs, rather than a single point estimate from the best-supported demographic model, as input values. For comparison, we also ran an additional set of 100 simulations using the point $N_e$ estimates obtained from the best-supported demographic model based on the WGS data.

### Whole-genome analyses

**Laboratory methods.** A representative selection of 20 northern elephant seals that passed stringent quality control (ten animals with helminth infection, three with bacterial infection, four with trauma and three with malnutrition) were subjected to WGS, together with 20 southern elephant seals. The DNA samples were measured photometrically using a NanoDrop One instrument (Thermo Fisher Scientific) to determine purity. DNA quality was determined by capillary electrophoresis using the Fragment Analyzer and DNF-464 HS Large Fragment 50 kb Kit (Agilent Technologies) and the final specific DNA concentration was determined using the fluorometric Qubit dsDNA BR assay (Thermo Fisher Scientific). Library preparation was performed according to the manufacturer's protocol using the Illumina DNA PCR-Free Prep, Tagmentation (Illumina) with a total input of >300 ng per sample. Libraries were normalized to 2 nM, pooled and sequenced on a NovaSeq 6000 (Illumina) with a read setup of 2 × 151 bp.

**Variant calling.** The genotyping of the WGS data from the northern and southern elephant seals was based on the GATK best practice recommendations[105]. The reference genome for the whole-genome analysis was the *M. angustirostris* genome, available via NCBI (RefSeq identification code GCF_021288785.2). Before genotyping, scaffolds smaller than 1 kb were removed. A subset of the analysis was also repeated with the *M. leonina* genome (RefSeq identification code GCF_011800145.1) as a reference. The genotyping included the conversion of the raw sequencing data into ubam format for the assignment of read groups and the marking of Illumina adaptor sequences. Subsequently, the data were mapped to the reference genome using BWA and duplicated reads were marked. Using the GATK tool HaplotypeCaller[106], genotype likelihoods were obtained for each sample and the final genotypes were called jointly on all samples using the GATK tool GenomicsDBImport. A threshold-based hard filtering of the raw genotypes was applied based on the metrics QD (<7.5), FS (>17.5), MQ (<55.0), SOR (>3.0), MQRankSum (<−0.5, >0.5) and ReadPosRankSum (<−2.25, >2.25). These thresholds were determined visually following the Broad Institute's recommendations on the hard filtering of germline short variants (https://gatk.broadinstitute.org/hc/en-us/articles/360035890471). Subsequent to the genotyping with GATK, missing genotypes were reformatted from the Broad Institute's notation (GT:0/0,DP:0) to

the standard vcf representation (GT:./.) using the BCFtools plugin +setGT[107] to prevent missing data from being erroneously interpreted as homozygote genotypes in the downstream analysis.

**Quality control.** The quality of the resequencing data was assessed with FastQC and interspecific contamination was ruled out with FastQScreen[108]. Mapping success was monitored with bamcov (https://github.com/fbreitwieser/bamcov) and BamTools[109]. To further rule out intraspecific contamination, a combination of VCFtools, GATK VariantsToTable and custom R scripts were used to check for allelic imbalance within the called SNPs for each individual. MultiQC[110] was used to monitor the quality control procedure and bundle individual reports.

**Genetic diversity.** To compare SNP densities between northern and southern elephant seals, VCFtools was used to create two subsets of genotypes, comprising only samples of one of the two species, respectively. These subsets were then filtered for a minor allele count of >1 to remove any invariant SNPs. Then, for both subsets, VCFtools was used to estimate SNP densities within non-overlapping 100 kb windows along the genome. For the estimation of nucleotide diversity ($\pi$), the genotyping step was repeated from the GenotypeGVCFs step onward, now including the flag --include-non-variant-sites true to also include invariant sites (the subsequent steps were unchanged). Based on this dataset, the Python scripts within the GitHub repository genomics_general (https://github.com/simonhmartin/genomics_general) were used to compute $\pi$ within 100 kb sliding windows with 25 kb increments. Additionally, individual heterozygosity was summarized within 1 Mb sliding widows with 250 kb increments using a combination of VCFtools and custom R scripts.

**ROH calling.** ROHs were called in both elephant seal species using BCFtools[111] and PLINK[85]. The BCFtools approach was implemented using the default parameters for the species subsets of the genotypes separately. For the PLINK analysis, a broad parameter space was explored by varying the input parameters over all possible combinations of --homozyg-window-snp 50, --homozyg-snp 100, --homozyg-kb [1000, 10], --homozyg-gap [1000, 50], --homozyg-density 50, --homozyg-window-missing [5, 20], --homozyg-het [1000, 0, 2] and --homozyg-window-het [1, 3] (a total of 48 parameter combinations). To evaluate the accuracy of the ROH calling, the resulting ROHs were then compared with patterns of genome-wide heterozygosity at the individual level using a combination of VCFtools and custom R scripts. Based on the close resemblance between the ROHs called by BCFtools and regions of low heterozygosity, the BCFtools results were favoured over the PLINK results for further analysis. A conservative minimum ROH length threshold of 1 Mb was then applied to facilitate comparisons with previous studies[112,113].

**Genomic mutation loads.** To identify functionally relevant SNPs in the genomes of the northern and southern elephant seals, genotypes within coding sequences were annotated using SnpEff[68]. For this analysis, we obtained the northern elephant seal genome annotation from NCBI (GCF_021288785.2_ASM2128878v3_genomic.gtf.gz) and extracted protein and coding sequences using the tool gff3_to_fasta from the GFF3toolkit (https://github.com/NAL-i5K/GFF3toolkit). A custom SnpEff database was then created using the SnpEff build command and the VCF file with the genotypes was annotated using the ann command with the flags -no-downstream, -no-intergenic, -no-intron, -no-upstream and -no-utr. Variants categorized as high-impact or loss-of-function variants were classified as putatively harmful mutations and included in the load scoring.

Next, we assigned ancestral alleles using a cactus[114] alignment of the two elephant seal genomes and the Weddell seal genome (RefSeq identification code GCF_000349705.1). Pairwise nucleotide differences

between the *M. angustirostris* genome and the inferred shared ancestor of the three seal species were exported from the alignment using halSnps[114]. The VCF file containing the genotypes was then further annotated with the inferred ancestral alleles using the vcf-annotate tool from VCFtools. Lastly, Jvarkit[115] was used in combination with a custom Java script to recode the genotypes based on their ancestral state (with 0 being ancestral and 1 being derived).

Finally, genomic mutation load estimates were obtained for all individuals using SnpSift[116]. For the inbreeding load, variants with putatively harmful mutations for the derived allele in the heterozygous state were tallied; for the segregating drift load, variants with putatively harmful mutations that were homozygous for the derived allele were scored. To estimate the magnitude of the drift load of each species, the genotypes were subsetted to include only SNPs that were invariant within each respective species. Then, for each subset, the number of SNPs that were fixed for the derived allele and that were classified as putatively harmful mutations was tallied using a combination of custom R scripts and SnpSift.

### Non-Wright–Fisher simulations

To investigate how close the northern elephant seal came to extinction and explore the probable impact of deleterious mutations on the recovery of the population, we implemented non-Wright–Fisher simulations in SLiM3. We modelled overlapping generations and implemented age-dependent mortality so that most females produced fewer than nine pups in total[117] and successful males died within a couple of years of first reproduction[118]. Mortality probabilities for the first four years of life were set to 0.632, 0.294, 0.253 and 0.160, respectively, according to Le Boeuf et al.[119]. Reproduction was implemented according to a harem-style system in which a small proportion of males reproduce with numerous females, each of which produce a single offspring per year. Le Boeuf[118] reported that the majority of copulations at Año Nuevo in California were undertaken by the five most active males, which accounted for less than 5% of the male population. We therefore set the proportion of reproducing males to 5% to account for the additional contributions of small numbers of opportunistic males with very low reproductive success. The age at first reproduction was set to 4 years for females, as this is the most common age at primiparity in female northern elephant seals[117], and 6 years for males[118]. The effect of deleterious mutations was assumed to be constant through all age classes and for each sex and was implemented entirely through survival, meaning that the genetic load purely affected the probability of an individual surviving to the next simulation cycle.

Census population sizes were determined by setting the carrying capacity ($K$) through time to appropriate values in the simulations. Given that the contemporary northern elephant seal population consists of around 225,000 individuals[30], we set the post-bottleneck $K$ to 350,000 to allow our simulated population to reach a contemporary $N_c$ similar or greater than the empirical value. The $N_c$ estimate of Lowry et al.[30] was also used to derive the census-to-effective population size ratio, which we then used to convert our historical $N_e$ estimates from the demographic model into $N_c$ values, and set the corresponding carrying capacities accordingly. $N_c$ at the start of the simulation (that is, preceding post-glacial expansion) was set to 2,670 ($10 \times N_{eLGM}$) assuming an $N_e$-to-$N_c$ ratio of 0.1. This was implemented as a time-effective solution for reaching an equilibrium level of genetic diversity during a burn-in of 26,700 simulation cycles, during which we implemented random mating. To investigate the effect of bottleneck strength on extinction risk and post-bottleneck population recovery, we set the bottleneck $K$ to 50, 100, 250, 500 and 1,000 and ran 100 simulations for each value. These values were chosen to allow comparisons to be made between strongly and weakly bottlenecked populations. Then, separately for each bottleneck scenario, we quantified the extinction probability as the proportion of simulations in which the northern elephant seal population went extinct. In addition, we extracted $N_c$ and fitness values

one generation before the bottleneck, as well as for each generation after the bottleneck (that is, every nine years, assuming a northern elephant seal generation time of 8.7 years). Simulated post-bottleneck demographic recovery trajectories were then compared with empirical values obtained from Bartholomew and Hubbs[27], Le Boeuf and Bonnell[28], Stewart et al.[29] and Lowry et al.[30]

Finally, we ran a series of neutral models to test whether any demographic patterns emerging from the non-Wright–Fisher models described above could be attributed to the stochasticity associated with the population decrease imposed by the bottleneck, rather than to fitness effects deriving from deleterious mutations. To do so, we re-ran all of the non-Wright–Fisher simulations while suppressing the onset of deleterious mutations and keeping everything else unchanged. Five simulations were run for each value of $K$.

## Reporting summary

Further information on research design is available in the Nature Portfolio Reporting Summary linked to this article.

## Data availability

Fitness data, microsatellite genotypes and a table of the genomic mutation loads of the northern and southern elephant seal, broken down by gene, are available from figshare[120]. The RAD sequencing and whole-genome resequencing data are available via the Sequence Read Archive (https://www.ncbi.nlm.nih.gov/bioproject/PRJNA1039994/, accession number PRJNA1039994).

## Code availability

The code used to analyse the fitness data is available at https://github.com/rshuhuachen/inbreeding-elephant-seals.git (ref. 121). The code used for the demographic reconstruction and forward genetic simulations is available at https://github.com/DavidVendrami/NorthernElephantSeals (ref. 122). The code for the whole-genome resequencing analyses can be accessed via https://github.com/k-hench/elephant_seals (ref.123). The execution of the whole-genome resequencing analyses was managed with Snakemake[124] and the computing environments were controlled using apptainer containers (Singularity Developers, 2021) and conda environments.

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

## Acknowledgements

This work was funded by the German Research Foundation (DFG) Sequencing Costs in Projects scheme (project number 497640428 awarded to J.I.H.), a standard DFG grant (project number 454606304 awarded to J.I.H.), and the DFG Schwerpunktprogramm (priority programme) 1158, 'Antarctic Research with Comparative Investigations in Arctic Ice Areas' (project number 424119118 awarded to J.I.H. and project number 522417425 awarded to K.H.). It was also partly supported by the Sonderforschungsbereiche (SFB) TRR 212 (NC$^3$, project number 316099922 awarded to O. Krüger and project number 396774617 awarded to J.I.H.). This work was also supported by the DFG Research Infrastructure West German Genome Center (project number 407493903 awarded to P. Nürnberg and M. Nothnagel) as part of the Next Generation Sequencing Competence Network (project number 423957469 awarded to J. Schultze). Sample quality control, library preparation, sequencing and post-sequencing quality control were carried out at the production site of the West German Genome Center in Düsseldorf. Additional funding was provided by the National Science Foundation (project numbers HRD 2000211 and OPP 2146068 awarded to C.A.B.). Support for the article processing charge was granted by the DFG and the Open Access Publication Fund of Bielefeld University. We gratefully acknowledge the staff and volunteers of The Marine Mammal Center for care of the elephant seals during rehabilitation and for collecting and archiving the samples. We also thank A. Knight and F. Christaller and P. Viehöver for assistance with DNA extraction and microsatellite genotyping, as well as H. Morales for discussions about SLiM. We are also grateful to the Zentrum für Informations- und Medientechnologie—especially the high-performance computing team at Heinrich-Heine University—for computational support. The University of York Viking high-performance compute facility was also used for this project.

## Author contributions

J.I.H., F.M.D.G., M.A.S. and K.K.D. conceived of the study idea. F.M.D.G., M.E.G. and C.A.B. contributed samples and veterinary data. J.I.H. provided funding, performed the microsatellite genotyping, analysed the data and supervised the research. M.A.S. and K.K.D. made the RAD sequencing libraries and J.K. sequenced them. T.W., D.R. and K.K. performed the WGS. R.S.C., D.L.J.V. and K.H. analysed the data. D.L.J.V. implemented the forward genetic simulations with help from M.K. and K.K.D. J.I.H. drafted the manuscript with input from F.M.D.G., D.L.J.V., K.H., R.S.C., M.K., W.A. and K.K.D. All of the authors commented on and approved the final manuscript.

## Funding

## Competing interests

The authors declare no competing interests.

## Additional information

**Extended data** is available for this paper at https://doi.org/10.1038/s41559-024-02533-2.

**Correspondence and requests for materials** should be addressed to Joseph I. Hoffman.

[1]Department of Evolutionary Population Genetics, Faculty of Biology, Bielefeld University, Bielefeld, Germany. [2]Center for Biotechnology (CeBiTec), Faculty of Biology, Bielefeld University, Bielefeld, Germany. [3]Department of Animal Behaviour, Faculty of Biology, Bielefeld University, Bielefeld, Germany. [4]British Antarctic Survey, Cambridge, UK. [5]Joint Institute for Individualisation in a Changing Environment (JICE), Bielefeld University and University of Münster, Bielefeld, Germany. [6]Northwest Fisheries Science Center, National Marine Fisheries Service, National Oceanic and Atmospheric Administration, Seattle, WA, USA. [7]Department of Zoology, University of Cambridge, Cambridge, UK. [8]Department of Microbial Genomics and Biotechnology, CeBiTec, Faculty of Biology, Bielefeld University, Bielefeld, Germany. [9]Genomics and Transcriptomics Laboratory, Biologisch-Medizinisches Forschungszentrum, and West German Genome Center, Heinrich-Heine-Universität Düsseldorf, Düsseldorf, Germany. [10]Institute of Marine Sciences, University of California, Santa Cruz, Santa Cruz, CA, USA. [11]Department of Marine and Environmental Sciences, Hampton University, Hampton, VA, USA. [12]Karen C. Drayer Wildlife Health Center, University of California, Davis, Davis, CA, USA. [13]Department of Biology, University of York, York, UK. [14]Present address: Alan Turing Institute, British Library, London, UK. ✉e-mail: joseph.hoffman@uni-bielefeld.de

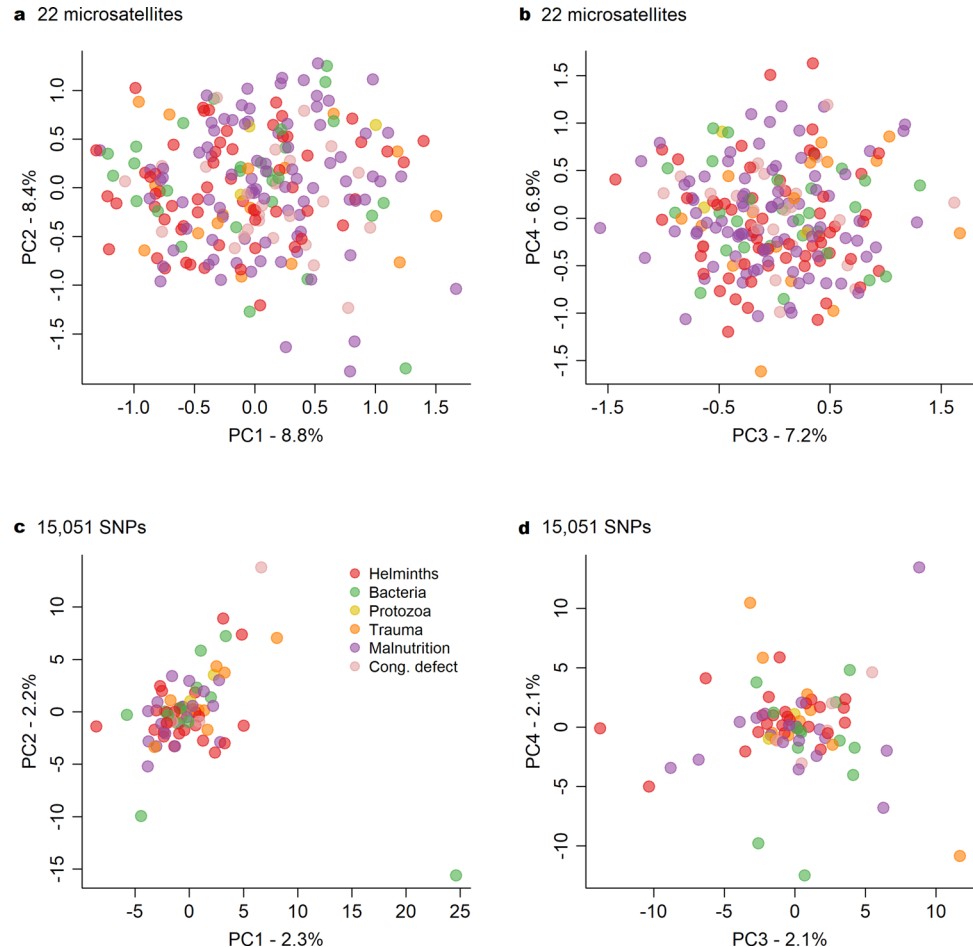

**Extended Data Fig. 1 | Scatterplots of individual variation in principal components.** Variation in principal components (PCs) one and two (panels **a** and **c**) and three and four (panels **b** and **d**) based on 22 microsatellites (panels **a** and **b**) and 15,051 SNPs (panels **c** and **d**). The amount of variance explained by each PC is shown on the respective axis labels.

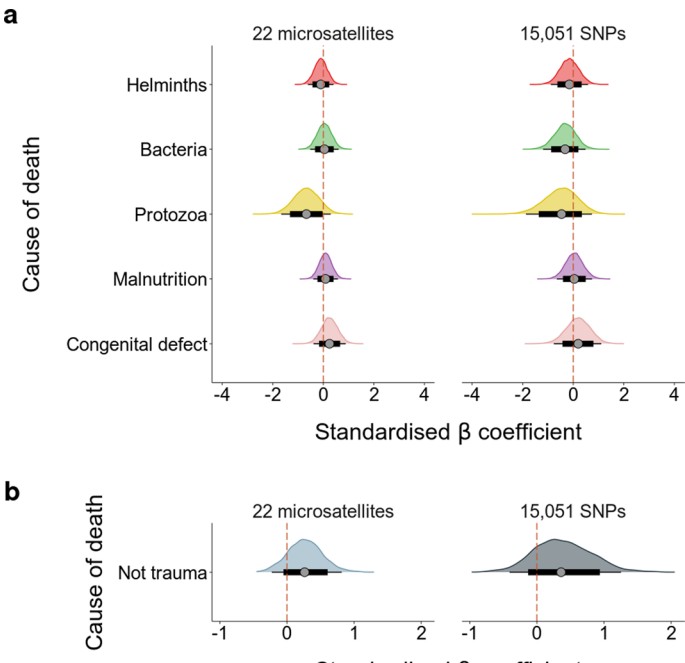

**Extended Data Fig. 2 | Results of Bayesian multinomial GLMMs where trauma was used as a reference category. a,** Posterior distributions of the standardized β coefficients of sMLH on binary classifications of the most likely causes of death. The same colour codes are used to represent the cause of death categories as in Fig. 1. **b,** Posterior distributions of the standardized β coefficients of sMLH on the binary classification: trauma versus all other causes of death combined. Grey points represent the posterior medians, black bars the 80% CIs and whiskers the 95% CIs.

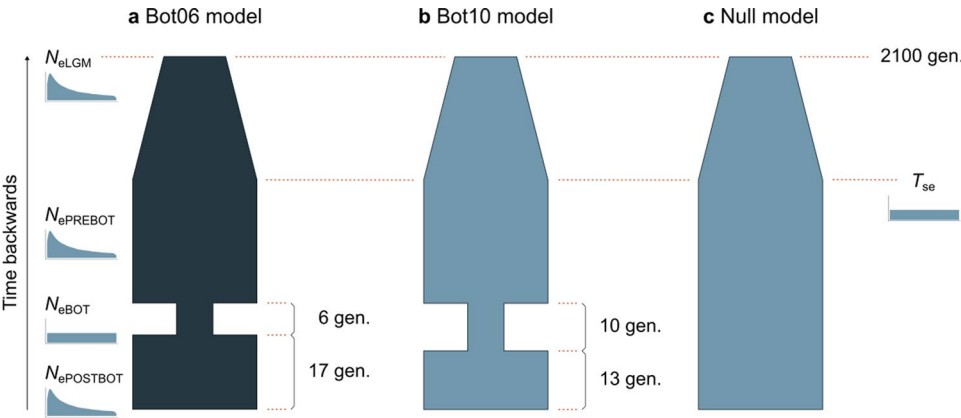

**Extended Data Fig. 3 | Schematic of three alternative demographic models.**
The demographic parameters used to define the models are shown together
with their prior distributions. In the models containing a bottleneck lasting six
generations (panel **a**, 'bot06 model') and a bottleneck lasting ten generations
(panel **b**, 'bot10 model'), five parameters were estimated (k = 5). We used log
uniform priors for $N_{eLGM}$ (the effective population size during the last glacial
maximum (LGM)), $N_{ePREBOT}$ (the effective population size before sealing) and

$N_{ePOSTBOT}$ (the effective population size in the current day), and uniform priors for
$N_{eBOT}$ (the effective population size during the bottleneck) and $T_{se}$ (the time of the
end of the LGM). The exact priors are specified in the Methods. The start time and
duration of the bottleneck were both fixed (to 23 generations ago and either six
or ten generations ago, respectively). The null model without a bottleneck (panel
**c**) included only the parameters $N_{eLGM}$, $N_{ePOSTBOT}$ and $T_{se}$ (k = 3) and used the same
priors as the bottleneck models.

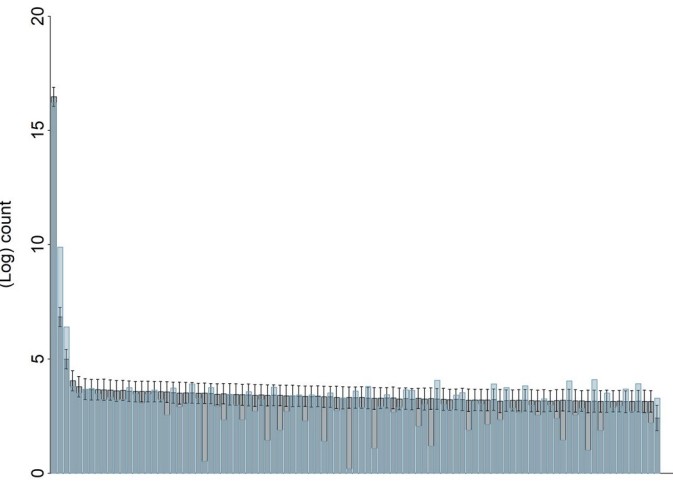

**Extended Data Fig. 4 | Observed and simulated site frequency spectra for the northern elephant seal.** The blue bars represent the empirical site frequency spectrum (SFS) based on RAD sequencing data from 96 individuals and the grey bars represent the mean SFS across 100 simulations based on the maximum likelihood parameter estimates from the best-supported model. The error bars represent the standard deviations around the means of the simulated values. The first bar on the left represents the number of monomorphic sites and the second bar represents the number of singletons, both of which were excluded from the demographic analysis as described in the Methods.

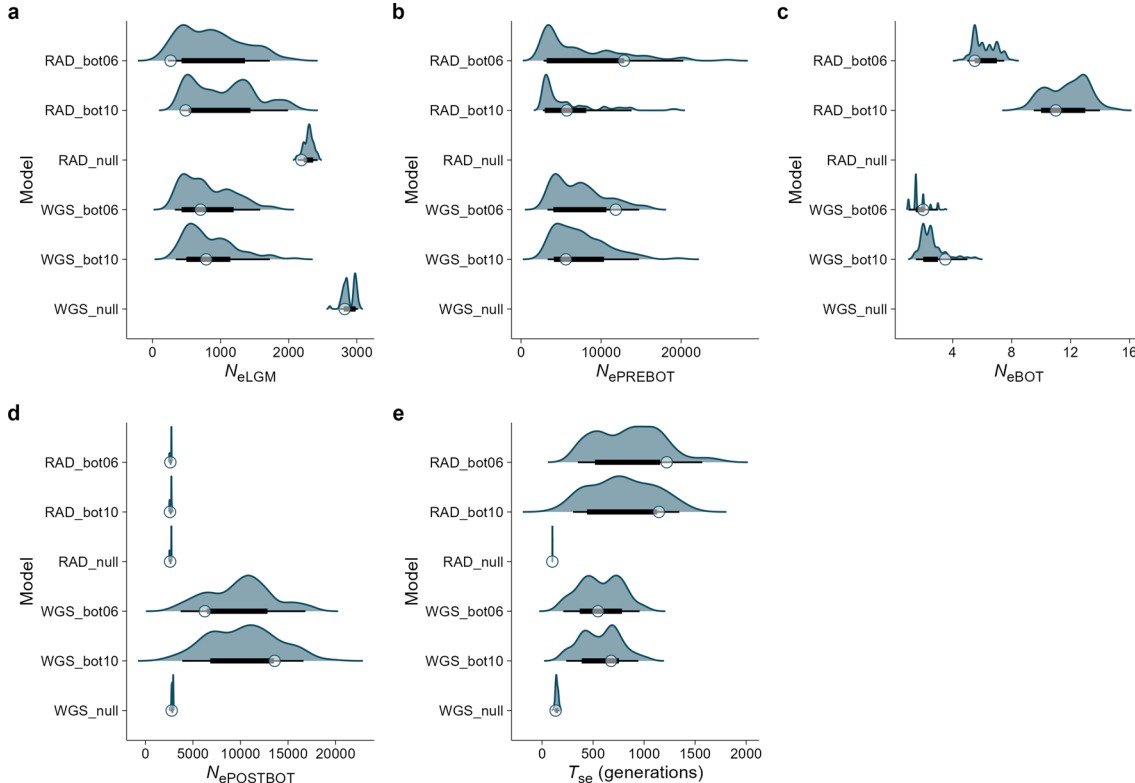

**Extended Data Fig. 5 | Parameter estimates obtained for the RAD sequencing (RAD) and whole genome sequencing (WGS) datasets.** Three alternative demographic models were evaluated for each dataset, the first including a bottleneck lasting for six generations (bot06), the second including a bottleneck lasting for ten generations (bot10) and a 'null' model that did not include a bottleneck. Shown are point estimates from the model with the best likelihood among 100 independent runs for each model (white points) and the distributions of 100 bootstrap replicates (blue shading: density distribution, thin lines 95% CIs, thick lines 66% CIs). **a**, $N_{eLGM}$: effective population size during the last glacial maximum (LGM). **b**, $N_{ePREBOT}$: effective population size before sealing. **c**, $N_{eBOT}$: effective population size during the bottleneck. **d**, $N_{ePOSTBOT}$: effective population size in the current day. **e**, $T_{se}$: time of the end of the LGM in generations ago.

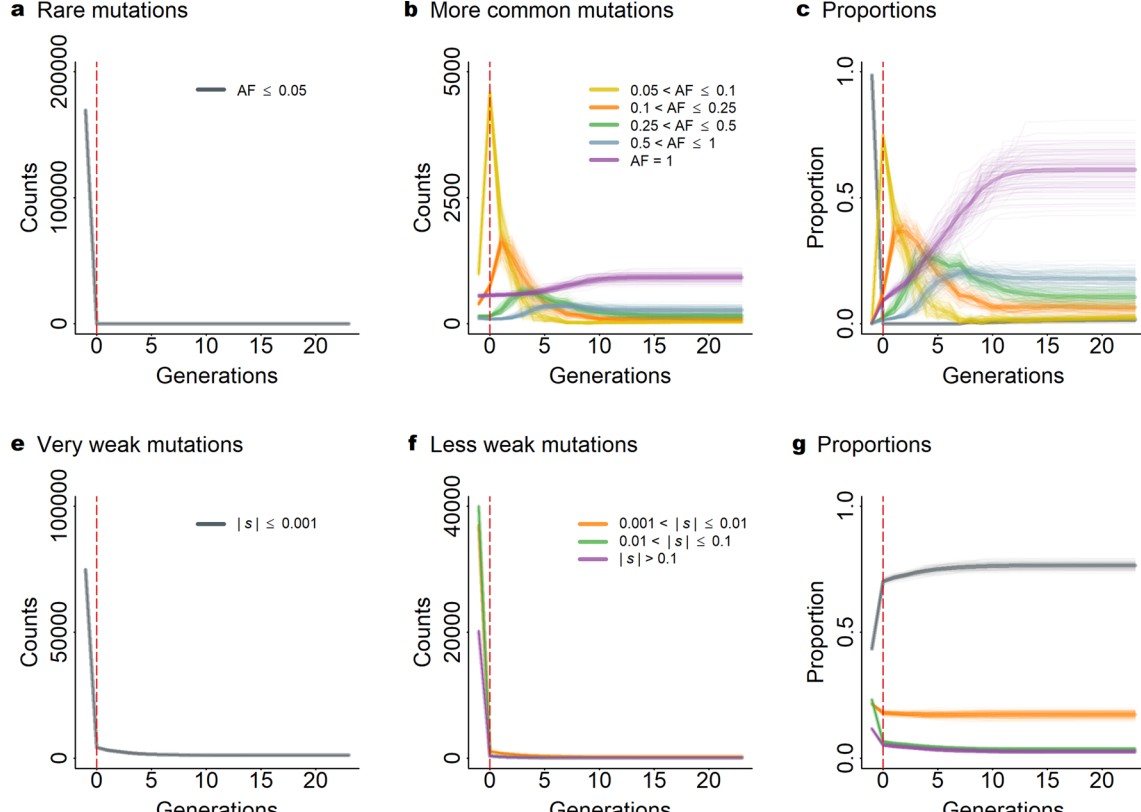

**Extended Data Fig. 6 | Counts and proportions of deleterious mutations in the simulated population shown over time (measured in generations, starting from one generation before the bottleneck until the present day), partitioned by allele frequency (AF) and effect size (*s*). a**, Counts of deleterious mutations with AF smaller than or equal to 5%. **b**, Counts of more common deleterious mutations (0.05 < AF ≤ 1) broken down into different AF classes as shown in the inset. **c**, The relative proportions of deleterious mutations belonging to the AF classes shown in the previous two panels. **d**, Counts of deleterious mutations with effect sizes less than or equal to 0.001. **e**, Counts of

deleterious mutations with larger effect sizes (0.001 < |*s*| > 0.1) broken down into different effect size classes as shown in the inset. **f**, The relative proportions of mutations with deleterious mutations belonging to the effect size classes shown in the previous two panels. Thick lines represent values averaged over 100 forward genetic Wright–Fisher simulations (thinner lines) based on point $N_e$ estimates from the best-supported demographic model derived from the RAD sequencing data (see Methods for details). The vertical dashed red lines indicate the onset of the bottleneck. Only simulated mutations that were present in the pre-bottleneck population were considered.

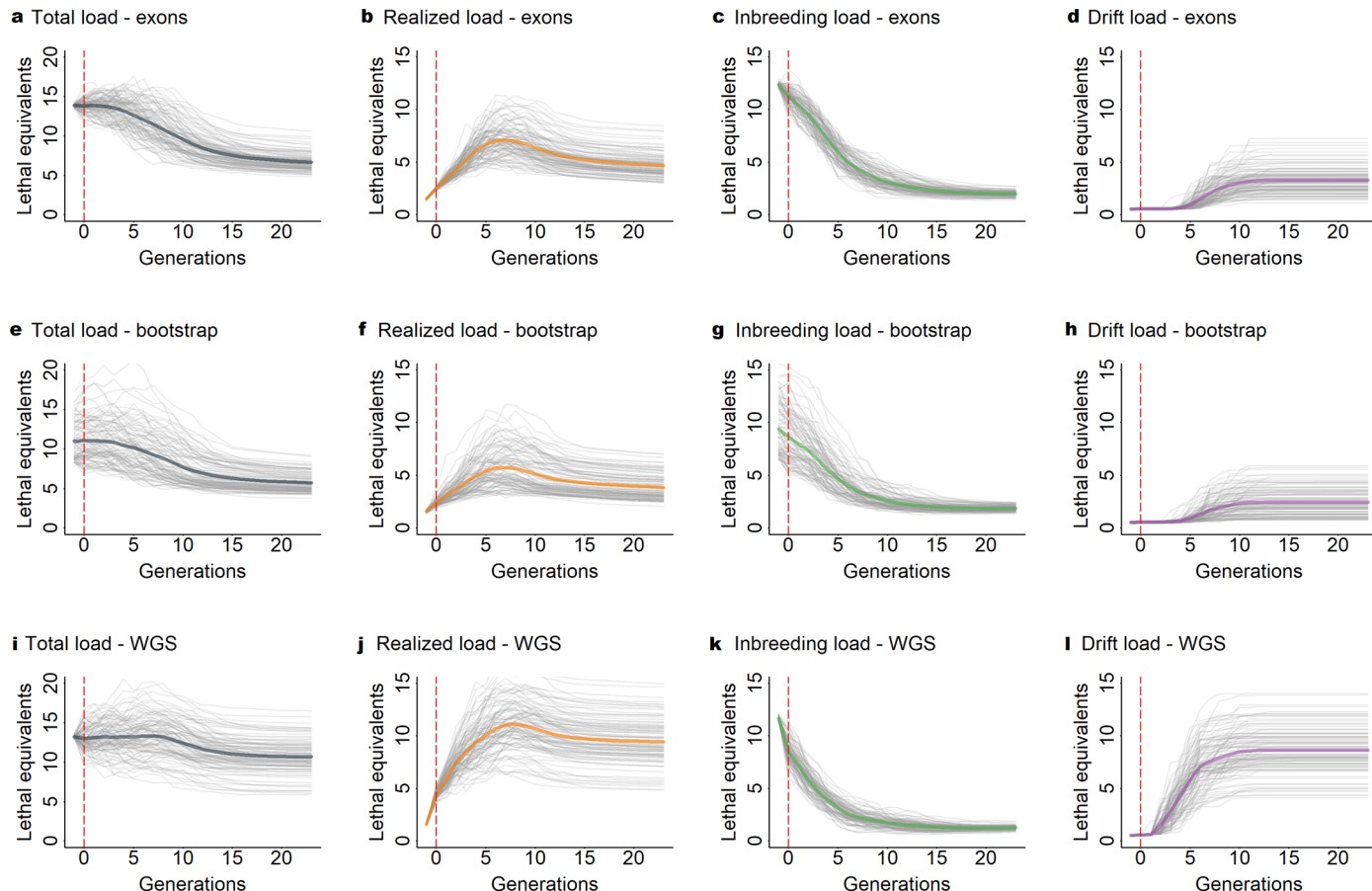

**Extended Data Fig. 7 | Sensitivity analysis of the results of the forward genetic Wright−Fisher simulations (see Methods for details).** Genetic loads in the simulated population are shown over time (measured in generations), starting from one generation before the bottleneck until the present day. Thick coloured lines represent values averaged over 100 forward genetic simulations (grey lines) and the vertical dashed red lines indicate the onset of the bottleneck. **a**−**d**, Genetic load estimates from simulations where deleterious mutations were allowed to arise only within exons, shown separately for the total load (**a**), the realized load (**b**), the inbreeding load (**c**) and the drift load (**d**). These models were based on point $N_e$ estimates from the best-supported demographic model derived from the RAD sequencing data. **e**−**h**, Genetic load estimates incorporating uncertainty from the demographic reconstruction, shown separately for the total load (**e**), the realized load (**f**), the inbreeding load (**g**) and the drift load (**h**). For this analysis, we used $N_e$ estimates from 100 bootstrap SFSs derived from the RAD sequencing data and allowed deleterious mutations to appear anywhere in the genome. **i**−**l**, Genetic load estimates from simulations based on the whole genome sequencing (WGS) data, shown separately for the total load (**i**), the realized load (**j**), the inbreeding load (**k**) and the drift load (**l**). For this analysis, we used point $N_e$ estimates from the best-supported demographic model derived from the WGS data and allowed deleterious mutations to appear anywhere in the genome.

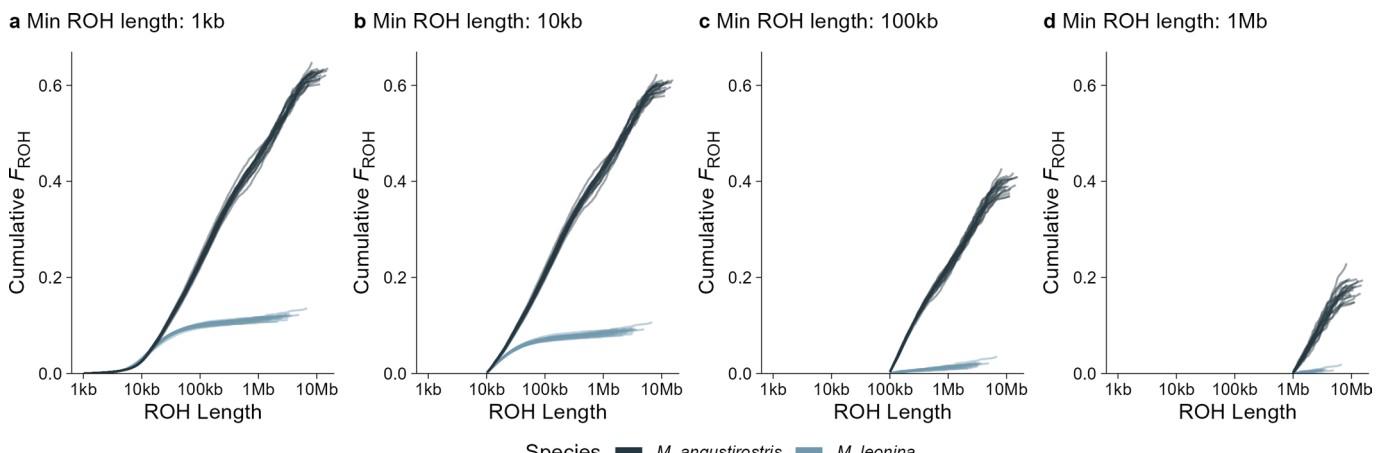

**Extended Data Fig. 8 | The influence of run of homozygosity (ROH) length threshold on genomic inbreeding estimates ($F_{ROH}$) in northern (*M. angustirostris*) and southern (*M. leonina*) elephant seals.** ROH length is shown on the x-axis on a log scale, while the y-axis shows the cumulative contribution to $F_{ROH}$ of ROH segments shorter than the indicated length. Each line represents a single individual, colour coded by species as shown in the legend. The minimum ROH length thresholds shown are: **a**, 1 kb. **b**, 10 kb. **c**, 100 kb and **d**, 1 Mb. For each individual, the rightmost value of $F_{ROH}$ corresponds to the overall genomic inbreeding score obtained when applying the respective ROH length threshold.

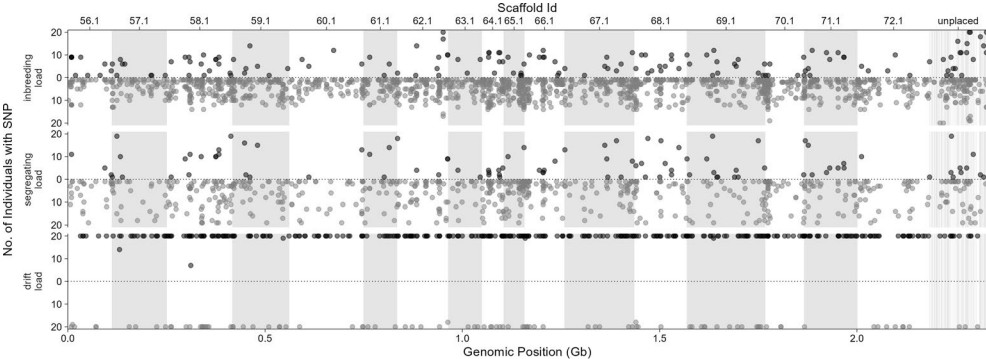

**Extended Data Fig. 9 | The genomic distribution of mutations contributing to the inbreeding load (upper plot), the segregating load (middle plot) and the drift load (bottom plot).** In each plot, the position of each SNP in the reference genome is shown along the x-axis and the number of individuals carrying each mutation is shown on the y-axis. The northern elephant seal is represented by dark grey points above the dotted line and the southern elephant seal is represented by light grey points below the dotted line, as shown in the legend. The white and grey shaded blocks in the background indicate the boundaries of consecutive scaffolds within the reference genome, with the number on top referring to the scaffold ID (where 'XX.X' stands for the chromosomal scaffold 'NC_0723XX.X').

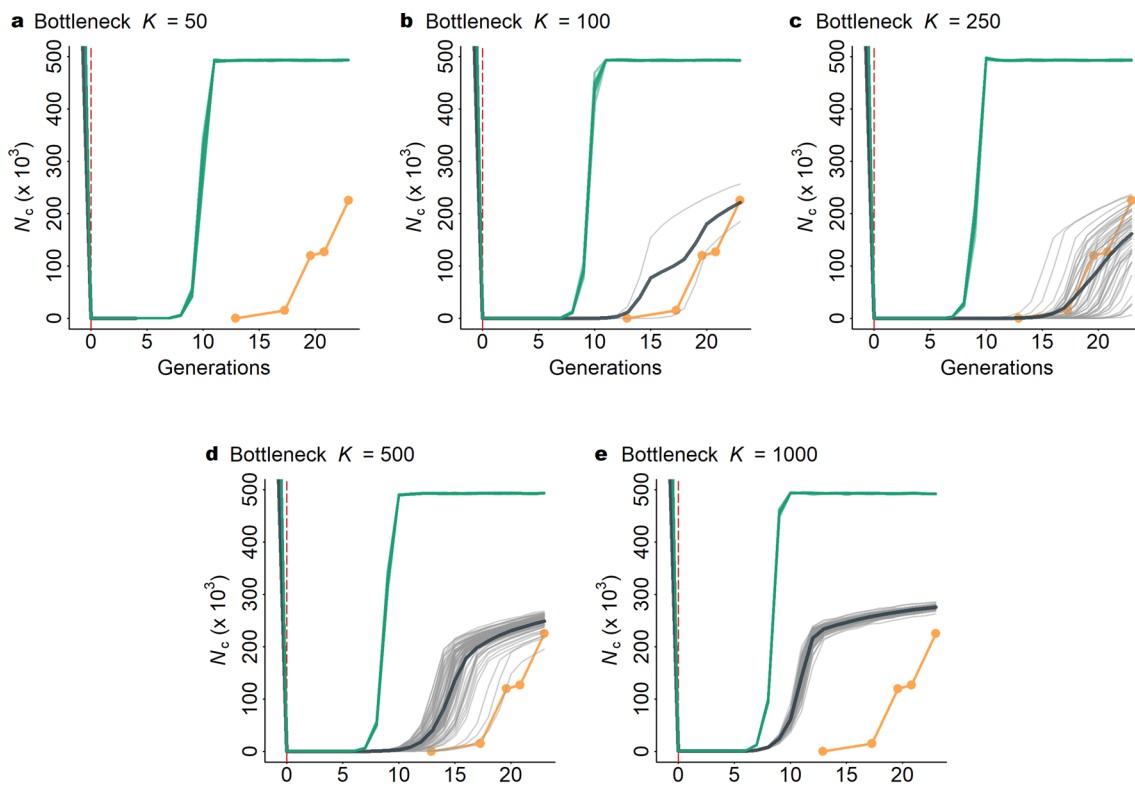

**Extended Data Fig. 10 | Results of non-Wright−Fisher -models of the northern elephant seal population omitting deleterious mutations ('neutral' models).** Results are shown for different sets of five non-Wright-Fisher neutral simulations that varied in the carrying capacity ($K$) of the population during the bottleneck. $N_c$ values were simulated over time (shown in generations), starting from one generation before the bottleneck to the present day, for: **a,** $K = 50$. **b,** $K = 100$. **c,** $K = 250$. **d,** $K = 500$ and **e,** $K = 1,000$. The thick green lines represent average

$N_c$ across the five simulations (light green lines). The results of the simulations including deleterious mutations (the same as in Fig. 5) are shown for comparison. The thick dark grey line represents the average $N_c$ of the simulations where the population did not go extinct (light grey lines). The orange points (joined by orange lines) indicate empirical $N_c$ estimates from Bartholomew & Hubbs[27], Le Boeuf & Bonnell[28], Stewart et al.[29] and Lowry et al.[30]. The vertical dashed red line indicates the onset of the bottleneck.

# Reporting Summary

## Statistics

For all statistical analyses, confirm that the following items are present in the figure legend, table legend, main text, or Methods section.

| n/a | Confirmed | |
|---|---|---|
| ☐ | ☒ | The exact sample size (*n*) for each experimental group/condition, given as a discrete number and unit of measurement |
| ☒ | ☐ | A statement on whether measurements were taken from distinct samples or whether the same sample was measured repeatedly |
| ☐ | ☒ | The statistical test(s) used AND whether they are one- or two-sided *Only common tests should be described solely by name; describe more complex techniques in the Methods section.* |
| ☐ | ☒ | A description of all covariates tested |
| ☐ | ☒ | A description of any assumptions or corrections, such as tests of normality and adjustment for multiple comparisons |
| ☐ | ☒ | A full description of the statistical parameters including central tendency (e.g. means) or other basic estimates (e.g. regression coefficient) AND variation (e.g. standard deviation) or associated estimates of uncertainty (e.g. confidence intervals) |
| ☐ | ☒ | For null hypothesis testing, the test statistic (e.g. *F*, *t*, *r*) with confidence intervals, effect sizes, degrees of freedom and *P* value noted *Give P values as exact values whenever suitable.* |
| ☐ | ☒ | For Bayesian analysis, information on the choice of priors and Markov chain Monte Carlo settings |
| ☒ | ☐ | For hierarchical and complex designs, identification of the appropriate level for tests and full reporting of outcomes |
| ☐ | ☒ | Estimates of effect sizes (e.g. Cohen's *d*, Pearson's *r*), indicating how they were calculated |

*Our web collection on statistics for biologists contains articles on many of the points above.*

## Software and code

Policy information about availability of computer code

| Data collection | All data are available via the provided links (see 'data availability') |
|---|---|
| Data analysis | All code are available via the provide links (see 'code availability') |

For manuscripts utilizing custom algorithms or software that are central to the research but not yet described in published literature, software must be made available to editors and reviewers. We strongly encourage code deposition in a community repository (e.g. GitHub). See the Nature Portfolio guidelines for submitting code & software for further information.

## Data

Policy information about availability of data

All manuscripts must include a data availability statement. This statement should provide the following information, where applicable:
- Accession codes, unique identifiers, or web links for publicly available datasets
- A description of any restrictions on data availability
- For clinical datasets or third party data, please ensure that the statement adheres to our policy

A data availability statement is provided

# Research involving human participants, their data, or biological material

Policy information about studies with [human participants or human data](). See also policy information about [sex, gender (identity/presentation), and sexual orientation]() and [race, ethnicity and racism]().

| | |
|---|---|
| Reporting on sex and gender | *Use the terms sex (biological attribute) and gender (shaped by social and cultural circumstances) carefully in order to avoid confusing both terms. Indicate if findings apply to only one sex or gender; describe whether sex and gender were considered in study design; whether sex and/or gender was determined based on self-reporting or assigned and methods used. Provide in the source data disaggregated sex and gender data, where this information has been collected, and if consent has been obtained for sharing of individual-level data; provide overall numbers in this Reporting Summary. Please state if this information has not been collected. Report sex- and gender-based analyses where performed, justify reasons for lack of sex- and gender-based analysis.* |
| Reporting on race, ethnicity, or other socially relevant groupings | *Please specify the socially constructed or socially relevant categorization variable(s) used in your manuscript and explain why they were used. Please note that such variables should not be used as proxies for other socially constructed/relevant variables (for example, race or ethnicity should not be used as a proxy for socioeconomic status). Provide clear definitions of the relevant terms used, how they were provided (by the participants/respondents, the researchers, or third parties), and the method(s) used to classify people into the different categories (e.g. self-report, census or administrative data, social media data, etc.) Please provide details about how you controlled for confounding variables in your analyses.* |
| Population characteristics | *Describe the covariate-relevant population characteristics of the human research participants (e.g. age, genotypic information, past and current diagnosis and treatment categories). If you filled out the behavioural & social sciences study design questions and have nothing to add here, write "See above."* |
| Recruitment | *Describe how participants were recruited. Outline any potential self-selection bias or other biases that may be present and how these are likely to impact results.* |
| Ethics oversight | *Identify the organization(s) that approved the study protocol.* |

Note that full information on the approval of the study protocol must also be provided in the manuscript.

# Field-specific reporting

Please select the one below that is the best fit for your research. If you are not sure, read the appropriate sections before making your selection.

☐ Life sciences ☐ Behavioural & social sciences ☒ Ecological, evolutionary & environmental sciences

For a reference copy of the document with all sections, see [nature.com/documents/nr-reporting-summary-flat.pdf]()

# Life sciences study design

All studies must disclose on these points even when the disclosure is negative.

| | |
|---|---|
| Sample size | *Describe how sample size was determined, detailing any statistical methods used to predetermine sample size OR if no sample-size calculation was performed, describe how sample sizes were chosen and provide a rationale for why these sample sizes are sufficient.* |
| Data exclusions | *Describe any data exclusions. If no data were excluded from the analyses, state so OR if data were excluded, describe the exclusions and the rationale behind them, indicating whether exclusion criteria were pre-established.* |
| Replication | *Describe the measures taken to verify the reproducibility of the experimental findings. If all attempts at replication were successful, confirm this OR if there are any findings that were not replicated or cannot be reproduced, note this and describe why.* |
| Randomization | *Describe how samples/organisms/participants were allocated into experimental groups. If allocation was not random, describe how covariates were controlled OR if this is not relevant to your study, explain why.* |
| Blinding | *Describe whether the investigators were blinded to group allocation during data collection and/or analysis. If blinding was not possible, describe why OR explain why blinding was not relevant to your study.* |

# Behavioural & social sciences study design

All studies must disclose on these points even when the disclosure is negative.

| | |
|---|---|
| Study description | *Briefly describe the study type including whether data are quantitative, qualitative, or mixed-methods (e.g. qualitative cross-sectional, quantitative experimental, mixed-methods case study).* |
| Research sample | *State the research sample (e.g. Harvard university undergraduates, villagers in rural India) and provide relevant demographic information (e.g. age, sex) and indicate whether the sample is representative. Provide a rationale for the study sample chosen. For studies involving existing datasets, please describe the dataset and source.* |

| | |
|---|---|
| Sampling strategy | *Describe the sampling procedure (e.g. random, snowball, stratified, convenience). Describe the statistical methods that were used to predetermine sample size OR if no sample-size calculation was performed, describe how sample sizes were chosen and provide a rationale for why these sample sizes are sufficient. For qualitative data, please indicate whether data saturation was considered, and what criteria were used to decide that no further sampling was needed.* |
| Data collection | *Provide details about the data collection procedure, including the instruments or devices used to record the data (e.g. pen and paper, computer, eye tracker, video or audio equipment) whether anyone was present besides the participant(s) and the researcher, and whether the researcher was blind to experimental condition and/or the study hypothesis during data collection.* |
| Timing | *Indicate the start and stop dates of data collection. If there is a gap between collection periods, state the dates for each sample cohort.* |
| Data exclusions | *If no data were excluded from the analyses, state so OR if data were excluded, provide the exact number of exclusions and the rationale behind them, indicating whether exclusion criteria were pre-established.* |
| Non-participation | *State how many participants dropped out/declined participation and the reason(s) given OR provide response rate OR state that no participants dropped out/declined participation.* |
| Randomization | *If participants were not allocated into experimental groups, state so OR describe how participants were allocated to groups, and if allocation was not random, describe how covariates were controlled.* |

# Ecological, evolutionary & environmental sciences study design

All studies must disclose on these points even when the disclosure is negative.

| | |
|---|---|
| Study description | Population genetics of elephant seals |
| Research sample | Northern and southern elephant seals |
| Sampling strategy | Described in manuscript |
| Data collection | Described in manuscript |
| Timing and spatial scale | Described in manuscript |
| Data exclusions | None |
| Reproducibility | All code and data have been made publically available so the study can be reproduced |
| Randomization | Not applicable |
| Blinding | Not applicable (no blinding performed) |

Did the study involve field work?   ☒ Yes   ☐ No

## Field work, collection and transport

| | |
|---|---|
| Field conditions | Sampling of random southern elephant seal pups on a study beach |
| Location | South Shetland Islands |
| Access & import/export | All performed under permit as described in the manuscript |
| Disturbance | Minimal |

# Reporting for specific materials, systems and methods

We require information from authors about some types of materials, experimental systems and methods used in many studies. Here, indicate whether each material, system or method listed is relevant to your study. If you are not sure if a list item applies to your research, read the appropriate section before selecting a response.

## Materials & experimental systems

| n/a | Involved in the study |
|---|---|
| ☒ ☐ | Antibodies |
| ☒ ☐ | Eukaryotic cell lines |
| ☒ ☐ | Palaeontology and archaeology |
| ☐ ☒ | Animals and other organisms |
| ☒ ☐ | Clinical data |
| ☒ ☐ | Dual use research of concern |
| ☒ ☐ | Plants |

## Methods

| n/a | Involved in the study |
|---|---|
| ☒ ☐ | ChIP-seq |
| ☒ ☐ | Flow cytometry |
| ☒ ☐ | MRI-based neuroimaging |

# Antibodies

| Antibodies used | Describe all antibodies used in the study; as applicable, provide supplier name, catalog number, clone name, and lot number. |
|---|---|
| Validation | Describe the validation of each primary antibody for the species and application, noting any validation statements on the manufacturer's website, relevant citations, antibody profiles in online databases, or data provided in the manuscript. |

# Eukaryotic cell lines

Policy information about cell lines and Sex and Gender in Research

| Cell line source(s) | State the source of each cell line used and the sex of all primary cell lines and cells derived from human participants or vertebrate models. |
|---|---|
| Authentication | Describe the authentication procedures for each cell line used OR declare that none of the cell lines used were authenticated. |
| Mycoplasma contamination | Confirm that all cell lines tested negative for mycoplasma contamination OR describe the results of the testing for mycoplasma contamination OR declare that the cell lines were not tested for mycoplasma contamination. |
| Commonly misidentified lines (See ICLAC register) | Name any commonly misidentified cell lines used in the study and provide a rationale for their use. |

# Palaeontology and Archaeology

| Specimen provenance | Provide provenance information for specimens and describe permits that were obtained for the work (including the name of the issuing authority, the date of issue, and any identifying information). Permits should encompass collection and, where applicable, export. |
|---|---|
| Specimen deposition | Indicate where the specimens have been deposited to permit free access by other researchers. |
| Dating methods | If new dates are provided, describe how they were obtained (e.g. collection, storage, sample pretreatment and measurement), where they were obtained (i.e. lab name), the calibration program and the protocol for quality assurance OR state that no new dates are provided. |

☐ Tick this box to confirm that the raw and calibrated dates are available in the paper or in Supplementary Information.

| Ethics oversight | Identify the organization(s) that approved or provided guidance on the study protocol, OR state that no ethical approval or guidance was required and explain why not. |
|---|---|

Note that full information on the approval of the study protocol must also be provided in the manuscript.

# Animals and other research organisms

Policy information about studies involving animals; ARRIVE guidelines recommended for reporting animal research, and Sex and Gender in Research

| Laboratory animals | No |
|---|---|
| Wild animals | Yes, elephant seals (as described in the manuscript) |
| Reporting on sex | Data on the sex of the animals are provided |
| Field-collected samples | Yes, southern elephant seals from the South Shetland Islands |

| Ethics oversight | Yes, all work was performed with appropriate ethical approval |
|---|---|

Note that full information on the approval of the study protocol must also be provided in the manuscript.

# Clinical data

Policy information about clinical studies
All manuscripts should comply with the ICMJE guidelines for publication of clinical research and a completed CONSORT checklist must be included with all submissions.

| Clinical trial registration | *Provide the trial registration number from ClinicalTrials.gov or an equivalent agency.* |
|---|---|
| Study protocol | *Note where the full trial protocol can be accessed OR if not available, explain why.* |
| Data collection | *Describe the settings and locales of data collection, noting the time periods of recruitment and data collection.* |
| Outcomes | *Describe how you pre-defined primary and secondary outcome measures and how you assessed these measures.* |

# Dual use research of concern

Policy information about dual use research of concern

## Hazards

Could the accidental, deliberate or reckless misuse of agents or technologies generated in the work, or the application of information presented in the manuscript, pose a threat to:

No | Yes
☐ ☐ Public health
☐ ☐ National security
☐ ☐ Crops and/or livestock
☐ ☐ Ecosystems
☐ ☐ Any other significant area

## Experiments of concern

Does the work involve any of these experiments of concern:

No | Yes
☐ ☐ Demonstrate how to render a vaccine ineffective
☐ ☐ Confer resistance to therapeutically useful antibiotics or antiviral agents
☐ ☐ Enhance the virulence of a pathogen or render a nonpathogen virulent
☐ ☐ Increase transmissibility of a pathogen
☐ ☐ Alter the host range of a pathogen
☐ ☐ Enable evasion of diagnostic/detection modalities
☐ ☐ Enable the weaponization of a biological agent or toxin
☐ ☐ Any other potentially harmful combination of experiments and agents

# Plants

| Seed stocks | *Report on the source of all seed stocks or other plant material used. If applicable, state the seed stock centre and catalogue number. If plant specimens were collected from the field, describe the collection location, date and sampling procedures.* |
|---|---|
| Novel plant genotypes | *Describe the methods by which all novel plant genotypes were produced. This includes those generated by transgenic approaches, gene editing, chemical/radiation-based mutagenesis and hybridization. For transgenic lines, describe the transformation method, the number of independent lines analyzed and the generation upon which experiments were performed. For gene-edited lines, describe the editor used, the endogenous sequence targeted for editing, the targeting guide RNA sequence (if applicable) and how the editor was applied.* |
| Authentication | *Describe any authentication procedures for each seed stock used or novel genotype generated. Describe any experiments used to assess the effect of a mutation and, where applicable, how potential secondary effects (e.g. second site T-DNA insertions, mosiacism, off-target gene editing) were examined.* |

# ChIP-seq

## Data deposition

☐ Confirm that both raw and final processed data have been deposited in a public database such as GEO.

☐ Confirm that you have deposited or provided access to graph files (e.g. BED files) for the called peaks.

Data access links
*May remain private before publication.*
> *For "Initial submission" or "Revised version" documents, provide reviewer access links. For your "Final submission" document, provide a link to the deposited data.*

Files in database submission
> *Provide a list of all files available in the database submission.*

Genome browser session
(e.g. UCSC)
> *Provide a link to an anonymized genome browser session for "Initial submission" and "Revised version" documents only, to enable peer review. Write "no longer applicable" for "Final submission" documents.*

## Methodology

Replicates
> *Describe the experimental replicates, specifying number, type and replicate agreement.*

Sequencing depth
> *Describe the sequencing depth for each experiment, providing the total number of reads, uniquely mapped reads, length of reads and whether they were paired- or single-end.*

Antibodies
> *Describe the antibodies used for the ChIP-seq experiments; as applicable, provide supplier name, catalog number, clone name, and lot number.*

Peak calling parameters
> *Specify the command line program and parameters used for read mapping and peak calling, including the ChIP, control and index files used.*

Data quality
> *Describe the methods used to ensure data quality in full detail, including how many peaks are at FDR 5% and above 5-fold enrichment.*

Software
> *Describe the software used to collect and analyze the ChIP-seq data. For custom code that has been deposited into a community repository, provide accession details.*

# Flow Cytometry

## Plots

Confirm that:

☐ The axis labels state the marker and fluorochrome used (e.g. CD4-FITC).

☐ The axis scales are clearly visible. Include numbers along axes only for bottom left plot of group (a 'group' is an analysis of identical markers).

☐ All plots are contour plots with outliers or pseudocolor plots.

☐ A numerical value for number of cells or percentage (with statistics) is provided.

## Methodology

Sample preparation
> *Describe the sample preparation, detailing the biological source of the cells and any tissue processing steps used.*

Instrument
> *Identify the instrument used for data collection, specifying make and model number.*

Software
> *Describe the software used to collect and analyze the flow cytometry data. For custom code that has been deposited into a community repository, provide accession details.*

Cell population abundance
> *Describe the abundance of the relevant cell populations within post-sort fractions, providing details on the purity of the samples and how it was determined.*

Gating strategy
> *Describe the gating strategy used for all relevant experiments, specifying the preliminary FSC/SSC gates of the starting cell population, indicating where boundaries between "positive" and "negative" staining cell populations are defined.*

☐ Tick this box to confirm that a figure exemplifying the gating strategy is provided in the Supplementary Information.

# Magnetic resonance imaging

## Experimental design

Design type
> *Indicate task or resting state; event-related or block design.*

| Design specifications | *Specify the number of blocks, trials or experimental units per session and/or subject, and specify the length of each trial or block (if trials are blocked) and interval between trials.* |
|---|---|
| Behavioral performance measures | *State number and/or type of variables recorded (e.g. correct button press, response time) and what statistics were used to establish that the subjects were performing the task as expected (e.g. mean, range, and/or standard deviation across subjects).* |

## Acquisition

| Imaging type(s) | *Specify: functional, structural, diffusion, perfusion.* |
|---|---|
| Field strength | *Specify in Tesla* |
| Sequence & imaging parameters | *Specify the pulse sequence type (gradient echo, spin echo, etc.), imaging type (EPI, spiral, etc.), field of view, matrix size, slice thickness, orientation and TE/TR/flip angle.* |
| Area of acquisition | *State whether a whole brain scan was used OR define the area of acquisition, describing how the region was determined.* |

Diffusion MRI ☐ Used   ☐ Not used

## Preprocessing

| Preprocessing software | *Provide detail on software version and revision number and on specific parameters (model/functions, brain extraction, segmentation, smoothing kernel size, etc.).* |
|---|---|
| Normalization | *If data were normalized/standardized, describe the approach(es): specify linear or non-linear and define image types used for transformation OR indicate that data were not normalized and explain rationale for lack of normalization.* |
| Normalization template | *Describe the template used for normalization/transformation, specifying subject space or group standardized space (e.g. original Talairach, MNI305, ICBM152) OR indicate that the data were not normalized.* |
| Noise and artifact removal | *Describe your procedure(s) for artifact and structured noise removal, specifying motion parameters, tissue signals and physiological signals (heart rate, respiration).* |
| Volume censoring | *Define your software and/or method and criteria for volume censoring, and state the extent of such censoring.* |

## Statistical modeling & inference

| Model type and settings | *Specify type (mass univariate, multivariate, RSA, predictive, etc.) and describe essential details of the model at the first and second levels (e.g. fixed, random or mixed effects; drift or auto-correlation).* |
|---|---|
| Effect(s) tested | *Define precise effect in terms of the task or stimulus conditions instead of psychological concepts and indicate whether ANOVA or factorial designs were used.* |

Specify type of analysis: ☐ Whole brain   ☐ ROI-based   ☐ Both

| Statistic type for inference<br>(See Eklund et al. 2016) | *Specify voxel-wise or cluster-wise and report all relevant parameters for cluster-wise methods.* |
|---|---|
| Correction | *Describe the type of correction and how it is obtained for multiple comparisons (e.g. FWE, FDR, permutation or Monte Carlo).* |

## Models & analysis

| n/a | Involved in the study |
|---|---|
| ☐ | ☐ Functional and/or effective connectivity |
| ☐ | ☐ Graph analysis |
| ☐ | ☐ Multivariate modeling or predictive analysis |

| Functional and/or effective connectivity | *Report the measures of dependence used and the model details (e.g. Pearson correlation, partial correlation, mutual information).* |
|---|---|
| Graph analysis | *Report the dependent variable and connectivity measure, specifying weighted graph or binarized graph, subject- or group-level, and the global and/or node summaries used (e.g. clustering coefficient, efficiency, etc.).* |
| Multivariate modeling and predictive analysis | *Specify independent variables, features extraction and dimension reduction, model, training and evaluation metrics.* |

