## [Peer Review File · Nature Ecology & Evolution]

Peer Review Information

Journal: Nature Ecology & Evolution

Manuscript Title: Genomic and fitness consequences of a near-extinction event in the northern elephant seal

Corresponding author name(s): Joseph Hoffman

Editorial Notes:

Reviewer Comments & Decisions:

Decision Letter, initial version:

1st March 2024

Dear Professor Hoffman,

Thank you for submitting your Article entitled "Genomic and fitness consequences of a near-extinction event in the northern elephant seal" for consideration. I regret to inform you that after careful consideration and discussion with my editorial colleagues, we have decided that we cannot consider it for publication in *Nature Ecology & Evolution*.

As you may know, we decline a substantial proportion of manuscripts without sending them to reviewers, so that they may be sent elsewhere without delay. In such cases, even if reviewers were to certify the manuscript as technically correct, we do not believe that it represents a development of sufficient importance to warrant publication in *Nature Ecology & Evolution*. These editorial judgements are based on such considerations as the degree of advance provided, the breadth of potential interest to researchers and timeliness.

In this case, we do not feel that your paper has matched our criteria for further consideration. We therefore feel that the paper would find a more suitable outlet in another journal.

Please be assured that this editorial decision does not represent a criticism of the quality of your work, nor are we questioning its value to others working in this area. We hope that you will rapidly receive a more favourable response elsewhere.

Although we cannot offer to publish your manuscript, we believe the editors at our sister journal, *Communications Biology*, will find it interesting and recommend you transfer it there; see below for more information about the journal and the benefits of transferring your manuscript.

I am sorry that we cannot respond more positively on this occasion.

[REDACTED]

2** *Communications Biology* is a selective Nature Portfolio title publishing Open Access research that brings new insight in all areas of the biological sciences. Additional journal metrics and information can be found here. Their editors prioritise good author service, fast peer review (in 2021, the median time to decision after first review was 40 days), and are happy to answer any questions you may have (commsbio@nature.com). The journal has an Impact Factor of 6.548, a CiteScore of 6.0 and a Scimago quartile ranking of Q1.

Please note that *Communications Biology* is a fully open-access journal and an article processing charge will apply to any papers accepted for publication. Our open access pages contain information about article processing charges, open access funding, and advice and support from Springer Nature.

**Although we cannot offer to publish your manuscript, I suggest that you consider *Communications Biology* as a suitable venue for this work. To transfer your manuscript, please use our manuscript transfer portal. You will not have to re-supply manuscript metadata and files, unless you wish to make modifications. For more information, please see our manuscript transfer FAQ page.

**For Nature Research general information and news for authors, see <http://npg.nature.com/authors>.

Decision Letter, first revision:

10th May 2024

Dear Joe,

Your manuscript entitled "Genomic and fitness consequences of a near-extinction event in the northern elephant seal" has now been seen by three reviewers, whose comments are attached. The reviewers have raised a number of concerns which will need to be addressed before we can offer publication in *Nature Ecology & Evolution*. We will therefore need to see your responses to the criticisms raised and to some editorial concerns, along with a revised manuscript, before we can reach a final decision regarding publication.

2We therefore invite you to revise your manuscript taking into account all reviewer and editor comments. Please highlight all changes in the manuscript text file [OPTIONAL: in Microsoft Word format].

- * Include a “Response to reviewers” document detailing, point-by-point, how you addressed each reviewer comment. If no action was taken to address a point, you must provide a compelling argument. This response will be sent back to the reviewers along with the revised manuscript.
- * If you have not done so already please begin to revise your manuscript so that it conforms to our Article format instructions at <http://www.nature.com/natecolevol/info/final-submission>. Refer also to any guidelines provided in this letter.
- * Include a revised version of any required reporting checklist. It will be available to referees (and, potentially, statisticians) to aid in their evaluation if the manuscript goes back for peer review. A revised checklist is essential for re-review of the paper.

[REDACTED]

Nature Ecology & Evolution is committed to improving transparency in authorship. As part of our efforts

in this direction, we are now requesting that all authors identified as 'corresponding author' on published papers create and link their Open Researcher and Contributor Identifier (ORCID) with their account on the Manuscript Tracking System (MTS), prior to acceptance. ORCID helps the scientific community achieve unambiguous attribution of all scholarly contributions. You can create and link your ORCID from the home page of the MTS by clicking on 'Modify my Springer Nature account'. For more information please visit please visit www.springernature.com/orcid.

[REDACTED]

Reviewer expertise:

Reviewer #1: population genetics, inbreeding, genetic load and purging

Reviewer #2: conservation and population genetics, marine mammals

Reviewer #3: conservation and population genetics of endangered animals

Reviewers' comments:

Reviewer #1 (Remarks to the Author):

This manuscript studies the genomic and fitness implications of the demographic history in the remarkable case of the northern elephant seal, a species recovering from the brink of extinction. I think this work makes a valuable contribution to conservation genetics.

First, it uses genetic markers (microsatellite loci and SNPs) to shows that the data do not allow to detect a significant association between individual heterozygosity and fitness traits, which is interpreted as a consequence of drift and genetic purging which could be ascribed to drift and purging (as well as to

4limited experimental power).

Second, a demographic reconstruction is obtained and used to simulate the evolution of the fitness genetic load and its hidden and expressed components. The results of this section are not groundbreaking, as the precise values obtained are dependent on the simulation parameter (including the mutational model), and the general pattern has already been described using analytical predictions and illustrated in different simulation analysis. The evolution of the inbreeding load represented in figure 3 is qualitatively consistent with the theoretical predictions obtained from the Inbreeding-Purging model (or, more exactly, with the more general Full Model where new mutation and non-purging selection are also considered) and also with other simulation results (See García-Dorado 2012). The same can be said regarding the evolution of the realized load, which is just a specular representation of the evolution of population mean fitness expected from inbreeding and purging (both caused by the bottleneck). The results reported here are useful as they correspond precisely to the demographic evolution inferred for the case study and to a reasonable mutational model. However, they should be presented as useful illustrations of the consequences of well-known theoretical predictions in this particular case.

Third, WGS is used to evaluate the genetic diversity and genomic inbreeding after bottlenecking, as well as the burden of putatively deleterious alleles, which are compared to those of a sister species that has not been through any substantial bottleneck. Both the genetic diversity and the burden of putatively deleterious alleles are substantially lower in the northern elephant seal than in its sister species. The reduction of the deleterious burden in the smaller population is a hallmark of genetic purging. The results are sound and valuable. They are consistent with previous findings, as this methodology has been used in several occasions up to now, generally revealing a reduction of the burden of deleterious derived alleles in populations that have undergone reductions in size of different magnitude. These previous evidences need to be discussed (Xue et al. 2015; Narashinhan et al. 2016, Grossen et al. 2020, Khan et al. 2021, Kleinman et al. 2022, analyzing data from gorilla, humans, ibex, lynx and tiger, respectively, and the list is not exhaustive). It is incomprehensible that none of these previous analysis have been even quoted, even more so when some of them have authors in common with this manuscript.

Finally, a simulation analysis is performed to evaluate the extinction risk under the demographic scenario inferred for the northern elephant seal. Despite all the uncertainties regarding the simulation parameter, the results are very illustrative and relevant to conservation. They show that the probability of extinction under bottlenecking was substantial, implying that this seal has been lucky while most species undergoing similar bottlenecks are more likely to be extinct. Results also show that the cause of

the extinction risk is mainly genetic (inbreeding depression, rather than demographic stochasticity). This implies that purging has substantially reduced that risk, that the species recovery has been possible due to genetic purging and that predictions ignoring purging are inadequate.

I think this manuscript is a clear and valuable contribution, although results need to be discussed in the context of the theoretical and experimental evidence. The other points raised above should also be addressed together with some punctual but conceptually relevant clarifications explained in the specific comments below.

Specific comments:

L. 62. Better “we find no evidence of inbreeding depression within the contemporary population”. This is to avoid confusion with the inbreeding depression of mean fitness compared with a reference population (for example, data from a sister species, as data previous to the bottleneck are not available).

L. 68. Rewrite: Conceptually, natural selection (including purging) does not cause fitness reduction. Variability for fitness causes natural selection, and natural selection causes an increase in fitness. Fitness inbreeding depression persisted for several generations. It prompted genetic purging, which first reduced inbreeding depression and then even caused some increase in fitness average.

L. 98: The inbreeding load is made just of the masked fraction of the effects: the inbreeding load is caused by partially recessive deleterious alleles, and is composed by the addition of the component of the deleterious effects that are masked in the heterozygous state.

L.180. Note that, even having statistical power to capture variability for inbreeding, there can be little statistical power to detect association between heterozygosity and the fitness traits. Particularly if fitness traits have important variation from other sources.

L. 238. There seems to be a misprint in the first CI. Anyway, these CI are known to be scarcely reliable as they do not capture relevant sources of error.

L 283-291: It is important to note here that a reduction in the total genetic load after a bottleneck can only be ascribed to genetic purging since, for additive deleterious alleles, natural selection is, at any time, more effective in larger populations.

L. 299: Note that the evolution of the realized load is the specular image of the evolution of mean fitness and behaves accordingly, as purging is expected to reduce inbreeding depression as inbreeding progresses and, at some point, revert previous fitness decline to some extent, as predicted by the Inbreeding-Purging model.

L. 328: Genes implied in oligogenic diseases are probably large effect deleterious, so that they should be purged more efficiently and with less stochasticity

L. 535 and elsewhere: Better “individual genomic mutation load”, just to distinguish it from the classical use of the term “mutation load” applying to the relative fitness reduction ascribed to segregating deleterious mutations. Alternatively, the term can be defined when used by the first time, with a comment on the classical meaning.

L. 390-393. It should be noted that this proxy is a very rough one. It is not just that it ignores the magnitudes of the deleterious effects. It also computes the inbreeding and realized load fractions by assuming that deleterious alleles are completely recessive. Even so, it seems to work to give a qualitative sensible picture.

L. 487: Better “disease within the contemporary ...” (see comment in the abstract)

L. 500. The literature cited here as reference for theoretical predictions only provides simulation results. For theoretical predictions of the evolution of the inbreeding and the expressed (or realized) load, including that due to fixation of deleterious alleles, García-Dorado 2012 can be quoted. For theoretical predictions of the corresponding evolution of the burden of the number of deleterious mutations Kleinman et al 2022 can be quoted.

L. 798. Maybe remove “the best”. I think this mutational model is a relatively sensible one, but there is enough uncertainty and polemics about how to integrate the available evidence as to avoid this expression.

Reviewer #2 (Remarks to the Author):

In this work Hoffman et al. take several lines of analysis to examine effects of the historic population bottleneck and subsequent recovery of northern elephant seals. Utilizing microsatellites, a set of reduced representation SNPs, and whole genome sequences the authors illustrate that the northern elephant seals examined did not show evidence of inbreeding depression, as well as reduced levels of ‘inbreeding load’ but higher ‘drift load’ compared to the southern elephant seals which had not

7experienced a bottleneck. These empirical results are paired with a variety of simulation analyses illustrating how the extreme bottleneck may have allowed northern elephant seals to purge deleterious variation, leading to the observed distributions of genetic load.

Overall, I found this piece to be very well written and the swath of analyses to be large but come together to tell a cohesive story.

My main concerns lie with the HFC analyses and if they are structured accurately. In addition, I found some of the analyses are not fully explained in the methods or results and discussion.

Regarding the HFC analyses, my main concern lies in the fact that all the individuals examined originate from a marine mammal recovery center and specifically only those individuals who passed away after intake. Thus, does this represent a 'random' set of individuals from which to characterize fitness and genomic diversity? This is partially addressed with having the 'trauma' group compared to those who had other conditions, but the sample size of this cohort is small. More context around this source of samples and possible biases for the analyses seems warranted.

Similarly, does it make sense to have each of these traits examined individually? Is the argument that the genetic architecture or selection pressure of each category would be different and therefore more/less likely to show an HFC? If so that context would be a good addition to the manuscript.

Regarding methods, on lines 157-159 it is mentioned that the samples did not show evidence of population structure, however details of these structure analyses are not given. Please elaborate on what program(s) and statistics were used.

In addition, on lines 371-373, ROH length threshold is not explained in the methods despite reference. Following this, on lines 876-879 it is stated that "a broad parameter space was explored" but no details are given. Can the authors expand on this in the text or point to relevant supplementary materials or github archives?

Reviewer #3 (Remarks to the Author):

I really enjoyed reading this very carefully written and very exiting paper.

Northern elephant seals are the textbook example of bottlenecks, and the early papers set the stage for conceptual, methodological and empirical explorations of the impacts of inbreeding on genetic variation. However, many of these investigations were unpowered and piecemeal. This paper brings many lines of investigation, together with high powered genomic data together to comprehensively investigate the impacts of precipitous population decline and inbreeding on this species.

Overall, I really like the framing of the questions, the sequential and detailed analyses, and the careful interpretation and validation of the results.

I have only one major comment is that this paper is truly population genetic, it talks about frequencies and categories of alleles, but since the authors have the genome, it would be nice to see where indeed this load is? What genes? What regions? I know this is preliminary, but still, adding one analysis on this would be nice.

As a minor comment, since so much seems to depend on the size and duration of the bottleneck, maybe the authors could run a couple of additional demographic history reconstruction approaches, like SMC++ and GONE.

I complement the authors on this amazing paper, and look forward to seeing it published.

*****END*****

Author Rebuttal, first revision:Rebuttal

We are grateful for the opportunity to submit a revision of our manuscript 'Genomic and fitness consequences of a near-extinction event in the northern elephant seal'. We also thank the three reviewers for their positive comments and suggestions, which have helped us to substantially improve our manuscript. In addition to the reviewer's suggestions, we have made three main further improvements to the manuscript:

- (i) We have included additional whole genome sequencing (WGS) data from ten more southern elephant seals. This increases our sample sizes of individuals for the WGS part of the paper to twenty for each species. This means that our sample sizes for this part of the study are now balanced, making the comparison of the diversity statistics for the two species easier to interpret. Please note that, as we did the genotyping of the two species jointly, this required us to repeat several downstream analyses for the northern elephant seals. However, the results did not change appreciably.
- (ii) We were made aware of the (for us) unexpected representation of missing data in the whole genome resequencing genotypes called by GATK. The version of GATK used for the study represents missing data in a non-standard way that can lead to missing genotypes being interpreted as homozygous for the reference allele and thereby inflate the estimates of homozygosity. To guard against this, we have added an additional step in our genotyping procedure, where we now explicitly restore the representation of missing data as per file specification for VCF files. Again, the results did not change appreciably.
- (iii) For the non-WF analysis, we increased the number of simulated populations per demographic scenario from 20 to 100 (see also Response 4). This allows us to more reliably quantify extinction probabilities and to better capture uncertainty in the resulting parameter estimates.

Reviewer #1 (Remarks to the Author):

This manuscript studies the genomic and fitness implications of the demographic history in the remarkable case of the northern elephant seal, a species recovering from the brink of extinction. I think this work makes a valuable contribution to conservation genetics.

Response 1: Thanks very much for the positive appraisal.

First, it uses genetic markers (microsatellite loci and SNPs) to show that the data do not allow to detect a significant association between individual heterozygosity and fitness traits, which is interpreted as a consequence of drift and genetic purging which could be ascribed to drift and purging (as well as to limited experimental power). Second, a demographic reconstruction is obtained and used to simulate the evolution of the fitness genetic load and its hidden and expressed components. The results of this section are not groundbreaking, as the precise values obtained are dependent on the simulation parameter (including the mutational model), and the general pattern has already been described using analytical predictions and illustrated in different simulation analysis. The evolution of the inbreeding load represented in figure 3 is qualitatively consistent with the theoretical predictions obtained from the Inbreeding-Purging model (or, more exactly, with the more general Full Model where new mutation and non-purging selection are also considered) and also with

other simulation results (See García-Dorado 2012). The same can be said regarding the evolution of the realized load, which is just a specular representation of the evolution of population mean fitness expected from inbreeding and purging (both caused by the bottleneck). The results reported here are useful as they correspond precisely to the demographic evolution inferred for the case study and to a reasonable mutational model. However, they should be presented as useful illustrations of the consequences of well-known theoretical predictions in this particular case.

Response 2: We concur about the agreement of the simulation results with theoretical expectations and now point this out explicitly and refer to García-Dorado 2012 in the relevant paragraph (lines 341-342) and in the introduction (line 116).

Third, WGS is used to evaluate the genetic diversity and genomic inbreeding after bottlenecks, as well as the burden of putatively deleterious alleles, which are compared to those of a sister species that has not been through any substantial bottleneck. Both the genetic diversity and the burden of putatively deleterious alleles are substantially lower in the northern elephant seal than in its sister species. The reduction of the deleterious burden in the smaller population is a hallmark of genetic purging. The results are sound and valuable. They are consistent with previous findings, as this methodology has been used in several occasions up to now, generally revealing a reduction of the burden of deleterious derived alleles in populations that have undergone reductions in size of different magnitude. These previous evidences need to be discussed (Xue et al. 2015; Narashinhan et al. 2016, Grossen et al. 2020, Khan et al. 2021, Kleinman et al. 2022, analyzing data from gorilla, humans, ibex, lynx and tiger, respectively, and the list is not exhaustive). It is incomprehensible that none of these previous analysis have been even quoted, even more so when some of them have authors in common with this manuscript.

Response 3: In response to this criticism, we have reworked the discussion surrounding the genomic mutation results, incorporating a new paragraph that discusses how our results compare to previous studies using similar methodologies (lines 481-494).

We have specifically cited the following additional papers:

- Robinson et al.¹
- Kleinman-Ruiz et al.²
- Khan et al. ³
- Robinson et al.⁴
- Van der Valk et al.⁵

Finally, a simulation analysis is performed to evaluate the extinction risk under the demographic scenario inferred for the northern elephant seal. Despite all the uncertainties regarding the simulation parameter, the results are very illustrative and relevant to conservation. They show that the probability of extinction under bottlenecks was substantial, implying that this seal has been lucky while most species undergoing similar bottlenecks are more likely to be extinct. Results also show that the cause of the extinction risk is mainly genetic (inbreeding depression, rather than demographic stochasticity). This implies that purging has substantially reduced that risk, that the species recovery has been possible due to genetic purging and that predictions ignoring purging are inadequate.

Response 4: After we submitted the manuscript, we updated our non-WF simulation analysis, making three improvements:

- (i) We increased the number of non-WF simulations from 20 to 100 in order to be able to better quantify extinction probabilities and uncertainty in the parameter estimates.

- (ii) Because the carrying capacity of the simulated populations (K) tells us little about the extent to which these populations were reduced, we have now included a detailed visualisation of temporal changes in the census population sizes (N_t) of the simulated populations (see updated Fig. 5).
- (iii) We discovered that our original scripts incorrectly specified pup and juvenile mortality due to typographic errors. We thoroughly re-checked all of our scripts and re-ran the analyses with the correct values.

The updated results show that the extinction probabilities did not change for the majority of the simulated scenarios, except for $K = 100$, which increased from 90% to $98\% \pm 2.8$ SD, and $K = 250$, which increased from 20% to $41\% \pm 9.8$ SD. The two surviving simulated populations for $K = 100$ were reduced to seven and 12 individuals respectively (Fig. 5c) while the surviving simulated populations for $K = 250$ were reduced to an average N_t of 22.7 ± 9 SD (Fig. 5d). Both the $K = 100$ and $K = 250$ scenarios produce demographic trajectories that are comparable to the empirical population recovery curve. Hence, our conclusions remain unaltered – the northern elephant seal population was likely reduced to somewhere in the order of 10 to 20 individuals and this decline was associated with a significant risk of extinction, somewhere between 41% and 98%. Furthermore, as indicated by the referees, the fitness trajectories of the simulated populations and comparisons to null models that do not include deleterious mutations confirm that extinction risk is strongly related to genetic effects.

I think this manuscript is a clear and valuable contribution, although results need to be discussed in the context of the theoretical and experimental evidence. The other points raised above should also be addressed together with some punctual but conceptually relevant clarifications explained in the specific comments below.

Response 5: We have incorporated the points raised above and below via changes throughout the text of the manuscript and by citing additional relevant literature (see also Responses 6-18).

Specific comments:

L. 62. Better “we find no evidence of inbreeding depression within the contemporary population”. This is to avoid confusion with the inbreeding depression of mean fitness compared with a reference population (for example, data from a sister species, as data previous to the bottleneck are not available).

Response 6: Thank you for the suggestion, which we have followed.

L. 68. Rewrite: Conceptually, natural selection (including purging) does not cause fitness reduction. Variability for fitness causes natural selection, and natural selection causes an increase in fitness. Fitness inbreeding depression persisted for several generations. It prompted genetic purging, which first reduced inbreeding depression and then even caused some increase in fitness average.

Response 7: Thank you for pointing this out. Given space constraints in the abstract, we elected to simplify this sentence by no longer referring to fitness and making a more direct connection between the purging of deleterious alleles and population recovery.

L. 98: The inbreeding load is made just of the masked fraction of the effects: the inbreeding load is caused by partially recessive deleterious alleles, and is composed by the addition of the component of the deleterious effects that are masked in the heterozygous state.

Response 8: We defined the inbreeding load according to Bertorelle et al.⁶. The only difference we can see between the definition we have used and that of the referee is that we stated that the inbreeding load is composed of the part of the fitness effects of segregating, partially recessive deleterious alleles that is in part masked in the heterozygous state. We have now removed 'in part' to align with the reviewer's definition. For improved clarity, the short definitions we used in the results are also accompanied by longer descriptions in the methods (lines 881-901) where we defined the inbreeding load as follows: 'The inbreeding load is the fraction of the total load that is masked in the heterozygous state. It is quantified by subtracting the realized load from the total load. This is the load component that determines inbreeding depression, as inbreeding unmasks the effects of deleterious mutations that are shielded from selection in the heterozygote state.'

L.180. Note that, even having statistical power to capture variability for inbreeding, there can be little statistical power to detect association between heterozygosity and the fitness traits. Particularly if fitness traits have important variation from other sources.

Response 9: Thank you for pointing this out. We have now explicitly acknowledged this point by including a referenced statement on lines 172-176.

L. 238. There seems to be a misprint in the first CI. Anyway, these CI are known to be scarcely reliable as they do not capture relevant sources of error.

Response 10: Thank you for spotting this mistake; we have now included the correct values.

L 283-291: It is important to note here that a reduction in the total genetic load after a bottleneck can only be ascribed to genetic purging since, for additive deleterious alleles, natural selection is, at any time, more effective in larger populations.

Response 11: We are not sure if the reviewer meant to suggest that genetic drift is unimportant to the dynamics of the load components through a bottleneck, but if so then we disagree: the loss of many rare alleles due to the strong genetic drift associated with population bottlenecks is a fundamental prediction of basic population genetics theory and is why deficits of rare alleles are a genetic hallmark of population bottlenecks (e.g.⁷). Most deleterious alleles have such weak fitness effects that they are essentially invisible to selection during a strong bottleneck and their rapid disappearance at the onset of a bottleneck must be explained in part by genetic drift. Of course, natural selection is also important, and that is a point we make consistently throughout the manuscript.

L. 299: Note that the evolution of the realized load is the specular image of the evolution of mean fitness and behaves accordingly, as purging is expected to reduce inbreeding depression as inbreeding progresses and, at some point, revert previous fitness decline to some extent, as predicted by the Inbreeding-Purging model.

Response 12: We agree that this result, and indeed all of the results described in that paragraph, are consistent with the theoretical predictions to which the reviewer refers. We now cite García-Dorado 2012 accordingly.

L. 328: Genes implied in oligogenic diseases are probably large effect deleterious, so that they should be purged more efficiently and with less stochasticity

Response 13: We agree and have changed the text accordingly.

L. 535 and elsewhere: Better “individual genomic mutation load”, just to distinguish it from the classical use of the term “mutation load” applying to the relative fitness reduction ascribed to segregating deleterious mutations. Alternatively, the term can be defined when used by the first time, with a comment on the classical meaning.

Response 14: Thanks for this suggestion – we now use the term ‘genomic mutation load’ as suggested. This is clearly defined and we explicitly describe how this differs from the classical meaning in lines 435–439.

L. 390–393. It should be noted that this proxy is a very rough one. It is not just that it ignores the magnitudes of the deleterious effects. It also computes the inbreeding and realized load fractions by assuming that deleterious alleles are completely recessive. Even so, it seems to work to give a qualitative sensible picture.

Response 15: This is correct and we have now clarified this point in the text (lines 438–439).

L. 487: Better “disease within the contemporary ...” (see comment in the abstract)

Response 16: we have changed this to ‘within the contemporary population’

L. 500. The literature cited here as reference for theoretical predictions only provides simulation results. For theoretical predictions of the evolution of the inbreeding and the expressed (or realized) load, including that due to fixation of deleterious alleles, Garcia-Dorado 2012 can be quoted. For theoretical predictions of the corresponding evolution of the burden of the number of deleterious mutations Kleinman et al 2022 can be quoted.

Response 17: Thank you for the suggestions. We have now included citations of these publications.

L. 798. Maybe remove “the best”. I think this mutational model is a relatively sensible one, but there is enough uncertainty and polemics about how to integrate the available evidence as to avoid this expression.

Response 18: We have done this.

Reviewer #2 (Remarks to the Author):

In this work Hoffman et al. take several lines of analysis to examine effects of the historic population bottleneck and subsequent recovery of northern elephant seals. Utilizing microsatellites, a set of reduced representation SNPs, and whole genome sequences the authors illustrate that the northern elephant seals examined did not show evidence of inbreeding depression, as well as reduced levels of ‘inbreeding load’ but higher ‘drift load’ compared to the southern elephant seals which had not experienced a bottleneck. These empirical results are paired with a variety of simulation analyses illustrating how the extreme bottleneck may have allowed northern elephant seals to purge deleterious variation, leading to the observed distributions of genetic load. Overall, I found this piece to be very well written and the swath of analyses to be large but come together to tell a cohesive story.

Response 19: Many thanks for the positive appraisal of our manuscript.

My main concerns lie with the HFC analyses and if they are structured accurately. In addition, I found some of the analyses are not fully explained in the methods or results and discussion.

Response 20: We have endeavoured to address this criticism by providing more context for our study and approach (See Responses 22 and 23), by more thoroughly

describing our population structure analyses (see response 24), and by providing more details of our ROH analysis (see response 24).

Regarding the HFC analyses, my main concern lies in the fact that all the individuals examined originate from a marine mammal recovery center and specifically only those individuals who passed away after intake. Thus, does this represent a 'random' set of individuals from which to characterize fitness and genomic diversity? This is partially addressed with having the 'trauma' group compared to those who had other conditions, but the sample size of this cohort is small. More context around this source of samples and possible biases for the analyses seems warranted.

Response 21: We do not believe that our study design should introduce strong biases for the following reasons:

- (i) The samples were assembled over several years and originated from a wide geographic area. Consequently, we believe our sample is representative of the wider population, rather than representing, for example, animals from a single breeding colony.
- (ii) Importantly, the trauma group should be random with respect to inbreeding, because there is no reason to expect inbreeding depression for accidental injuries such as boat strikes and bite wounds. In line with this, Acevedo-Whitehouse et al.⁸ did not find any evidence for California sea lions with trauma being inbred, which is consistent with our results for the northern elephant seal.
- (iii) A 'genetic bias' could potentially arise if certain alleles present in our sample were not representative of the larger population. In this regard, there is a general tendency for animals with higher heterozygosity to have lower disease risk^{8,9}. However, heterozygosity carries little information about which specific alleles are carried, so is unlikely on its own to create a genetic bias. By contrast, a few studies have reported trends where animals either carrying or lacking particular alleles at a locus differ in their disease susceptibility^{8,10}. If this were the case in elephant seals, a genetic bias might result from sampling sick animals. However, the bias would not be genome-wide, but instead would apply only to the particular gene and strongly linked regions, leaving the rest of the genome unaffected. Hence, such a bias, if present, would not be expected to result in a difference in genome-wide heterozygosity between the fitness categories.

We are grateful to the reviewer for raising this point and have written a new paragraph that openly discusses the potential for biases to arise from our sampling design (see lines 236-247).

Similarly, does it make sense to have each of these traits examined individually? Is the argument that the genetic architecture or selection pressure of each category would be different and therefore more/less likely to show an HFC? If so that context would be a good addition to the manuscript.

Response 22: We analysed the traits individually following Acevedo-Whitehouse et al.⁸, who conducted a similar study of California sea lions at the same seal rehabilitation centre. They found that average internal relatedness at 11 microsatellites differed significantly among six different categories of 'health problem', including some of the same categories used in our manuscript (specifically:

trauma, bacterial infection and helminth infection). Thus, our prior expectation was that the effect size of inbreeding might well vary among the categories, reflecting differences in genetic architecture and / or selection pressures as suggested by the referee. We have now included a more detailed justification of our approach in our revised manuscript (lines 196-199).

In response to the question 'does it make sense to have each of these traits examined individually', we believe the answer is 'yes' based on the arguments above. Nevertheless, we have now implemented an additional analysis that compares the trauma category with all of the other categories combined. Again, the 95% CIs of the posterior distributions of the standardized beta coefficients of sMLH again overlapped zero for both the microsatellite and SNP datasets, suggesting that there is no difference in inbreeding between individuals with ill health and otherwise healthy 'control' animals. We describe the results of this analysis on lines 211-216 and in a new panel of Extended Data Fig. 2.

Regarding methods, on lines 157-159 it is mentioned that the samples did not show evidence of population structure, however details of these structure analyses are not given. Please elaborate on what program(s) and statistics were used.

Response 23: Our initial inference of a lack of population structure was based on the results of principal component analyses (PCA) of the microsatellite and SNP datasets. The results, shown in Extended Data Fig 1, reveal no evidence of multiple genetic clusters, suggesting that our samples originate from a genetically homogenous population. Furthermore, there is no pattern of clustering by disease status.

The reviewer is correct in noting that we inadvertently did not include the methods (including software) we used for the PCA in the methods section and we are grateful for having this pointed out to us. We have now written a new, referenced, Methods section (lines 708-712). To further improve clarity, we also described the results more thoroughly on lines 158-163.

In addition, on lines 371-373, ROH length threshold is not explained in the methods despite reference. Following this, on lines 876-879 it is stated that "a broad parameter space was explored" but no details are given. Can the authors expand on this in the text or point to relevant supplementary materials or github archives?

Response 24: We thank the reviewer for pointing out these deficiencies in our description of the ROH calling. In response to this, we now justify the choice of our minimum length threshold mainly on the basis of comparability with similar studies. We also present (in Extended Data Fig. 8) and discuss the influence of the minimum ROH length threshold setting on the estimates of F_{ROH} (lines 422-426). To improve clarity, we have also expanded the Methods sub-heading "ROH calling", where we now explicitly list all of the parameter combinations explored in plink (lines 987-992). In addition to this, we have slightly restructured the code behind the ROH calling within the github repository to make the ROH calling parameters more accessible.

Reviewer #3 (Remarks to the Author):

I really enjoyed reading this very carefully written and very exiting paper.

Response 25: We're very glad to hear that!

Northern elephants seals are the textbook example of bottlenecks, and the early papers set the stage for conceptual, methodological and empirical explorations of the impacts of inbreeding on genetic variation. However, many of these investigations were unpowered and piecemeal. This paper bring many lines of investigation, together with high powered genomic data together to comprehensively investigate the impacts of precipitous population decline and inbreeding on this species.

Overall, I really like the framing of the questions, the sequential and detailed analyses, and the careful interpretation and validation of the results.

Response 26: Thanks again for the positive comments.

I have only one major comment is that this paper is truly population genetic, it talks about frequencies and categories of alleles, but since the authors have the genome, it would be nice to see where indeed this load is? What genes? What regions? I know this is preliminary, but still, adding one analyses on this would be nice.

Response 27: We thank the reviewer for their interest in our study system and our data. To address this point, we have expanded the Results and Discussion sub-heading “Genomic inbreeding and individual genomic mutation loads”, where we now discuss the distribution of the mutation load along the genome and summarise the load carried over all annotated genes (lines 465-479). To this means, we have added a new figure (Extended Data Fig. 9) that displays the distribution of all three load types along the genome in both elephant seal species. Additionally, we now include a table with detailed gene-specific load-tallies within the data submitted to figshare.

As a minor comment, since so much seems to depend on the size and duration of the bottleneck, maybe the authors could run a couple of additional demographic history reconstruction approaches, like SMC++ and GONE.

Response 28: We agree with the reviewer that size and duration of the bottleneck are important parameters in our forward genetic simulations. Because of this, we have already performed two extensive sensitivity analyses:

First, we explored the sensitivity of the demographic results to model parameters such as bottleneck duration and the type of data used for the demographic reconstruction (RAD sequencing versus WGS). The results, which are described in lines 274–287 and shown in Extended data fig. 5 and Supplementary table 4, show that the demographic models all converge on rather similar parameter estimates, especially for the key parameter N_{BOT} .

Second, we evaluated how uncertainty in our demographic parameter estimates influences the outcomes of the Wright-Fisher simulations. These analyses are described in lines 364–379 and the results are shown in Extended data fig. 7. They show that the inference of purging is highly robust to uncertainty in the parameter estimates.

Nevertheless, we welcome the reviewer's suggestion of running additional demographic reconstructions to confirm our parameter estimates. Unfortunately, however, SMC++ and GONE do not appear particularly well suited to our goals for the following reasons:

- (i) Neither approach allows the formulation and testing of specific demographic scenarios.

- (ii) Neither approach allows the bottleneck to be fixed to the specific timeframe during which it is known to have occurred.
- (iii) While approaches such as SMC++ can capture long-term changes in N , they are not well suited to inferring very recent changes in N (e.g. during the last 10–20 generations when the population recovery occurred).
- (iv) GONE produces a point N estimate using the Jorde-Ryman modification to the temporal method to account for the age structure of a population. This is problematic in our case because we do not have discrete, non-overlapping temporal samples and because the resulting N estimate will likely correspond more closely to the contemporary than the historical N (as it is based on allele frequency changes during the specific timeframe covered by the temporal sampling).

To allow comparison with our results from `fastsimcoal2`, we therefore decided to use a different software, `dadi`¹¹, to compare demographic models with different bottleneck durations and to infer effective population sizes. Notably, `dadi` relies on diffusion approximations, which is very different from the approach we implemented in `fastsimcoal2`. The results we obtained from `dadi` were comparable to those presented in our manuscript (see Table 1, below), with N_{BOT} being consistently in the order of 2–8 regardless of the method of inference and data type. Thus, our results and conclusions regarding the demographic history of the northern elephant seal were largely confirmed by `dadi`.

Table 1. Summary of the results of demographic reconstructions using `Fastsimcoal2` and `dadi` for the RAD sequencing and whole genome resequencing (WGS) datasets. 95% CIs are given in square brackets.

	RAD sequencing		WGS	
	<code>Fastsimcoal2</code>	<code>dadi</code>	<code>Fastsimcoal2</code>	<code>dadi</code>
N_{PREBOT}	12,856 [2,828 - 20,275]	20,019 [1,031 - 20,023]	11,836 [3297, 14763]	18,751 [7,144 - 19,844]
N_{BOT}	6 [5 - 7.5]	6 [4 - 8]	2 [1, 3]	8 [7 - 8]
N_{POSTBOT}	2,624 [2,506 - 2,773]	2,857 [2,232 - 5,163]	6,248 [3708, 16824]	1,597 [923 - 2,941]

While these results are reassuring, we do not believe that adding them will improve the manuscript for the following reasons:

- (i) The model we implemented with `dadi` is not strictly comparable to the `fastsimcoal` model because it assumes instantaneous growth from the LGM to the pre-bottleneck N .
- (ii) The `dadi` analysis represents a *post-hoc* confirmatory analysis that is built on top of our `fastsimcoal` analysis. This is because the ancestral population size that `dadi` uses to express any estimated N_e value and time-point is not estimated from the data. Rather, we defined this as the post-glacial N estimated by `fastsimcoal`.
- (iii) In the interests of brevity, we also believe that it would be better not to include the `dadi` results in the revised manuscript due to the lengthy

description that would be required to explain our implementation of the dadi analysis, and because we would also want to perform additional confirmatory and exploratory analyses that would require more space.

However, we have included a very short and straightforward additional analysis based on the loss of heterozygosity inferred from the comparison of the mean individual genome-wide heterozygosity of our contemporary samples with the mean individual genome-wide heterozygosity of the pre-bottleneck samples of Hoelzel et al.¹² Assuming that the latter estimate is an accurate reflection of the average pre-bottleneck heterozygosity of the northern elephant seal, the harmonic mean N_e since the onset of the bottleneck can be estimated by solving:

$$f_t = 1 - \left(1 - \left(\frac{1}{2N_e} \right) \right)^t$$

for N_e after setting t (the number of generations since the onset of intensive harvesting) to 23 and f_t (the proportional reduction in average heterozygosity in the post- versus pre-bottleneck population) to $0.00142 - 0.00018 / 0.00142 = 0.87$. This yields an estimated harmonic mean N_e of 5.83 for the northern elephant seal over the 23 generations since the onset of the bottleneck, further supporting the conclusion of extremely small N_e during the bottleneck.

We hope that these results will help to further reassure the referee that our results and inferences are quite robust.

I complement the authors on this amazing paper, and look forward to seeing it published.
Response 29: Thank you very much again for the helpful suggestions, which helped us to improve the manuscript.

References cited:

- 1 Robinson, J. A., Brown, C., Kim, B. Y., Lohmueller, K. E. & Wayne, R. K. Purging of strongly deleterious mutations explains long-term persistence and absence of inbreeding depression in island foxes. *Current Biology* 28, 3487–3494 (2018).
- 2 Kleinman-Ruiz, D. *et al.* Purging of deleterious burden in the endangered Iberian lynx. *Proceedings of the National Academy of Sciences* 119, e2110614119, doi:10.1073/pnas.2110614119 (2022).
- 3 Khan, A. *et al.* Genomic evidence for inbreeding depression and purging of deleterious genetic variation in Indian tigers. *Proceedings of the National Academy of Sciences* 118, e2023018118, doi:10.1073/pnas.2023018118 (2021).
- 4 Robinson, J. A. *et al.* The critically endangered vaquita is not doomed to extinction by inbreeding depression. *Science* 376, 635–639 (2022).
- 5 Van Der Valk, T., Díez-Del-Molino, D., Marques-Bonet, T., Guschanski, K. & Dalén, L. Historical genomes reveal the genomic consequences of recent population decline in eastern gorillas. *Current Biology* 29, 165–170, doi:10.2139/ssrn.3254908 (2019).
- 6 Bertorelle, G. *et al.* Genetic load: genomic estimates and applications in non-model animals. *Nature Reviews Genetics* 23, 492–503, doi:10.1038/s41576-022-00448-x (2022).

- 7 Luikart, G., Allendorf, F. W., Cornuet, J. M. & Sherwin, W. B. Distortion of allele frequency distributions provides a test for recent population bottlenecks. *Journal of Heredity* 89, 238–247, doi:10.1093/jhered/89.3.238 (1998).
- 8 Acevedo-Whitehouse, K., Gulland, F., Greig, D. & Amos, W. Inbreeding: Disease susceptibility in California sea lions. *Nature* 422, 35, doi:10.1038/422035a (2003).
- 9 MacDougall-Shackleton, E. A., Derryberry, E. P., Foufopoulos, J., Dobson, A. P. & Hahn, T. P. Parasite-mediated heterozygote advantage in an outbred songbird population. *Biology Letters* 1, 105–107, doi:10.1098/rsbl.2004.0264 (2005).
- 10 Garcia, A. A. *et al.* Association between major histocompatibility complex haplotypes and susceptibility of unvaccinated and vaccinated cattle to paratuberculosis. *Veterinary Immunology and Immunopathology* 265, 110677 (2023).
- 11 Gutenkunst, R., Hernandez, R., Williamson, S. & Bustamante, C. Diffusion Approximations for Demographic Inference: DaDi. *Nature Precedings*, doi:10.1038/npre.2010.4594.1 (2010).
- 12 Hoelzel, A. R. *et al.* Genomics of post-bottleneck recovery in the northern elephant seal. *Nature Ecology and Evolution* 8, 686–694, doi:10.1038/s41559-024-02337-4 (2024).

Decision Letter, second revision:

Our ref: NATECOLEVOL-24020507B

16th July 2024

Dear Dr. Hoffman,

Thank you for your patience as we've prepared the guidelines for final submission of your Nature Ecology & Evolution manuscript, "Genomic and fitness consequences of a near-extinction event in the northern elephant seal" (NATECOLEVOL-24020507B). Please carefully follow the step-by-step instructions provided in the attached file, and add a response in each row of the table to indicate the changes that you have made. Please also check and comment on any additional marked-up edits we have proposed within the text. Ensuring that each point is addressed will help to ensure that your revised manuscript can be swiftly handed over to our production team.

****We would like to start working on your revised paper, with all of the requested files and forms, as soon as possible (preferably within two weeks). Please get in contact with us immediately if you anticipate it taking more than two weeks to submit these revised files.****

In recognition of the time and expertise our reviewers provide to Nature Ecology & Evolution's editorial process, we would like to formally acknowledge their contribution to the external peer review of your manuscript entitled "Genomic and fitness consequences of a near-extinction event in the northern elephant seal". For those reviewers who give their assent, we will be publishing their names alongside the published article.

22Nature Ecology & Evolution offers a Transparent Peer Review option for new original research manuscripts submitted after December 1st, 2019. As part of this initiative, we encourage our authors to support increased transparency into the peer review process by agreeing to have the reviewer comments, author rebuttal letters, and editorial decision letters published as a Supplementary item. When you submit your final files please clearly state in your cover letter whether or not you would like to participate in this initiative. Please note that failure to state your preference will result in delays in accepting your manuscript for publication.

Cover suggestions

We welcome submissions of artwork for consideration for our cover. For more information, please see our guide for cover artwork.

Nature Ecology & Evolution has now transitioned to a unified Rights Collection system which will allow our Author Services team to quickly and easily collect the rights and permissions required to publish your work. Approximately 10 days after your paper is formally accepted, you will receive an email in providing you with a link to complete the grant of rights. If your paper is eligible for Open Access, our Author Services team will also be in touch regarding any additional information that may be required to arrange payment for your article.

Please note that *Nature Ecology & Evolution* is a Transformative Journal (TJ). Authors may publish their research with us through the traditional subscription access route or make their paper immediately open access through payment of an article-processing charge (APC). Authors will not be required to make a final decision about access to their article until it has been accepted. Find out more about Transformative Journals

Authors may need to take specific actions to achieve compliance with funder and institutional open access mandates. If your research is supported by a funder that requires immediate open access (e.g.

according to Plan S principles) then you should select the gold OA route, and we will direct you to the compliant route where possible. For authors selecting the subscription publication route, the journal's standard licensing terms will need to be accepted, including <https://www.nature.com/nature-portfolio/editorial-policies/self-archiving-and-license-to-publish>. Those licensing terms will supersede any other terms that the author or any third party may assert apply to any version of the manuscript.

[REDACTED]

[REDACTED]

Reviewer #1:

Remarks to the Author:

The authors have addressed the questions raised and I think the work constitutes a relevant contribution to the area of conservation genetics. Therefore, I advise publication of the manuscript. Below I clarify a few comments of my previous report that seem to have been difficult to interpret and that I think could still help to further improve the manuscript, in case the authors consider them in their final manuscript.

Abstract:

The abstract continues stating that selection reduces fitness (“selection against deleterious alleles persisted for several generations after the bottleneck, impeding the northern elephant seal’s demographic recovery”). I conceptually disagree with this view. When natural selection acts against deleterious alleles and, therefore, favors advantageous alleles, it does not reduce fitness. Natural selection is the consequence of reduced fitness, not its cause: In the generation when selection occurs, the variability for fitness (which implies some segregating load, i.e., some reduction in fitness below

maxim possible value) prompts natural selection which, in turn, is expected to increase future fitness. What threatens the population is the persistence of the deleterious alleles, not natural selection.

Comment to previous version line 98 (now 108):

What I meant here is that the inbreeding load is not “composed of segregating ... alleles that are in part masked”. First, it is not composed of alleles but of deleterious effects; second it is not composed by all the effect but just by the part of the effects that is masked.

Comment to previous version line 283-291 (now 376).

I do not discuss the role of drift modifying the frequency distribution of deleterious alleles. Drift is indeed expected to reduce the number of rare alleles after a bottleneck, but not the total count of derived alleles per genome including fixed alleles in the count, which is invariant to population size. Regarding natural selection against additive deleterious alleles, it is always less effective in smaller populations (or during bottlenecking), due to genetic drift. Only during purging the total load of derived alleles (segregating plus fixed) reduces faster in smaller population. Thus, a in the number of rare alleles is indeed a hallmark of population bottlenecks, but a reduction in total number of deleterious alleles per genome is a hallmark of genetic purging. I think this argument could be useful here.

Reviewer #2:

Remarks to the Author:

This is my second time reviewing the manuscript and I appreciate the lengths the authors went to address all the referee comments, expand the dataset, and address a potential error in the analytical pipeline. I do not have additional concerns at this time.

Reviewer #3:

Remarks to the Author:

I am satisfied with the revised version of this manuscript, and happy to recommend publication.

Author Rebuttal, second revision:

Point-by-point response to the reviewers' comments

Reviewer #1 (Remarks to the Author):

The authors have addressed the questions raised and I think the work constitutes a relevant contribution to the area of conservation genetics. Therefore, I advise publication of the manuscript. Below I clarify a few comments of my previous report that seem to have been difficult to interpret and that I think could still help to further improve the manuscript, in case the authors consider them in their final manuscript.

Response 1. Thank you for the further clarifications. We have accordingly made the changes described below in responses 2 and 3.

Abstract:

The abstract continues stating that selection reduces fitness ("selection against deleterious alleles persisted for several generations after the bottleneck, impeding the northern elephant seal's demographic recovery"). I conceptually disagree with this view. When natural selection acts against deleterious alleles and, therefore, favors advantageous alleles, it does not reduce fitness. Natural selection is the consequence of reduced fitness, not its cause. In the generation when selection occurs, the variability for fitness (which implies some segregating load, i.e., some reduction in fitness below maximum possible value) prompts natural selection which, in turn, is expected to increase future fitness. What threatens the population is the persistence of the deleterious alleles, not natural selection.

Response 2. We have replaced the sentence in the abstract with a more general formulation that does not refer to selection, fitness or population growth.

Comment to previous version line 98 (now 108):

What I meant here is that the inbreeding load is not "composed of segregating ... alleles that are in part masked". First, it is not composed of alleles but of deleterious effects, second it is not composed by all the effect but just by the part of the effects that is masked.

Response 3. We have altered the definition to avoid the formulation 'composed of segregating alleles'.

Comment to previous version line 283-291 (now 376):

I do not discuss the role of drift modifying the frequency distribution of deleterious alleles. Drift is indeed expected to reduce the number of rare alleles after a bottleneck, but not the total count of derived alleles per genome including fixed alleles in the count, which is invariant to population size. Regarding natural selection against additive deleterious alleles, it is always less effective in smaller populations (or during bottlenecking), due to genetic drift. Only during purging the total load of derived alleles (segregating plus fixed) reduces faster in smaller population. Thus, a reduction in the number of rare alleles is indeed a hallmark of population bottlenecks, but a reduction in total number of deleterious alleles per genome is a hallmark of genetic purging. I think this argument could be useful here.

Response 4. Thanks for the clarification; we're happy that we're on the same page regarding purging and drift, both of which affect deleterious allele frequencies in heavily bottlenecked populations. However, we do not feel that this argument is particularly helpful in the context of our paper, especially given that our simulations did not include neutral alleles, precluding an empirical comparison of rare-neutral versus rare-deleterious allele frequency changes.

Final Decision Letter:

7th August 2024

Dear Professor Hoffman,

We are pleased to inform you that your Article entitled "Genomic and fitness consequences of a near-extinction event in the northern elephant seal", has now been accepted for publication in *Nature Ecology & Evolution*.

Over the next few weeks, your paper will be copyedited to ensure that it conforms to *Nature Ecology and Evolution* style. Once your paper is typeset, you will receive an email with a link to choose the appropriate publishing options for your paper and our Author Services team will be in touch regarding any additional information that may be required

Due to the importance of these deadlines, we ask you please us know now whether you will be difficult to contact over the next month. If this is the case, we ask you provide us with the contact information (email, phone and fax) of someone who will be able to check the proofs on your behalf, and who will be available to address any last-minute problems . Once your paper has been scheduled for online publication, the Nature press office will be in touch to confirm the details.

Acceptance of your manuscript is conditional on all authors' agreement with our publication policies (see www.nature.com/authors/policies/index.html). In particular your manuscript must not be published elsewhere and there must be no announcement of the work to any media outlet until the publication date (the day on which it is uploaded onto our web site).

Please note that *Nature Ecology & Evolution* is a Transformative Journal (TJ). Authors may publish their research with us through the traditional subscription access route or make their paper immediately open access through payment of an article-processing charge (APC). Authors will not be required to make a final decision about access to their article until it has been accepted. Find out more about Transformative Journals

27Authors may need to take specific actions to achieve compliance with funder and institutional open access mandates. If your research is supported by a funder that requires immediate open access (e.g. according to Plan S principles) then you should select the gold OA route, and we will direct you to the compliant route where possible. For authors selecting the subscription publication route, the journal's standard licensing terms will need to be accepted, including <https://www.nature.com/nature-portfolio/editorial-policies/self-archiving-and-license-to-publish>. Those licensing terms will supersede any other terms that the author or any third party may assert apply to any version of the manuscript.

We welcome the submission of potential cover material (including a short caption of around 40 words) related to your manuscript; suggestions should be sent to Nature Ecology & Evolution as electronic files (the image should be 300 dpi at 210 x 297 mm in either TIFF or JPEG format). Please note that such pictures should be selected more for their aesthetic appeal than for their scientific content, and that colour images work better than black and white or grayscale images. Please do not try to design a cover with the Nature Ecology & Evolution logo etc., and please do not submit composites of images related to your work. I am sure you will understand that we cannot make any promise as to whether any of your suggestions might be selected for the cover of the journal.

To assist our authors in disseminating their research to the broader community, our SharedIt initiative provides you with a unique shareable link that will allow anyone (with or without a subscription) to read the published article. Recipients of the link with a subscription will also be able to download and print

the PDF.

You can generate the link yourself when you receive your article DOI by entering it here: <http://authors.springernature.com/share>.

[REDACTED]

P.S. Click on the following link if you would like to recommend Nature Ecology & Evolution to your librarian <http://www.nature.com/subscriptions/recommend.html#forms>

** Visit the Springer Nature Editorial and Publishing website at www.springernature.com/editorial-and-publishing-jobs for more information about our career opportunities. If you have any questions please click here.**